# Conditional Diffusion with Ordinal Regression: Longitudinal Data Generation for Neurodegenerative Disease Studies

**Hyuna Cho**[1], **Ziquan Wei**[2], **Seungjoo Lee**[1], **Tingting Dan**[2], **Guorong Wu**[2], **Won Hwa Kim**[1]
[1]POSTECH, South Korea
[2]University of North Carolina at Chapel Hill, USA

## Abstract

Modeling the progression of neurodegenerative diseases such as Alzheimer's disease (AD) is crucial for early detection and prevention given their irreversible nature. However, scarcity of longitudinal data and complex disease dynamics make the analysis highly challenging. Moreover, longitudinal samples often contain irregular and large intervals between subject visits, which underscore the necessity for advanced data generation techniques that can accurately simulate disease progression over time. In this regime, we propose a novel conditional generative model for synthesizing longitudinal sequences and present its application to neurodegenerative disease data generation conditioned on multiple time-dependent ordinal factors, such as age and disease severity. Our method sequentially generates continuous data by bridging gaps between sparse data points with a diffusion model, ensuring a realistic representation of disease progression. The synthetic data are curated to integrate both cohort-level and individual-specific characteristics, where the cohort-level representations are modeled with an ordinal regression to capture longitudinally monotonic behavior. Extensive experiments on four AD biomarkers validate the superiority of our method over nine baseline approaches, highlighting its potential to be applied to a variety of longitudinal data generation.

## 1 Introduction

**Motivation & Background.** Modeling progressive changes in the brain is vital for early diagnosis of neurodegenerative diseases and effective treatment planning (Paulsen et al., 2013). However, the task is highly challenging due to the scarcity of samples. That is, the data are collected from longitudinal visits from individuals over a long period, and they become significantly limited and imbalanced due to increased mortality as the disease worsens. Moreover, it requires sophisticated neuroimaging techniques such as positron emission tomography (PET) and magnetic resonance imaging (MRI) to observe the progression *in-vivo*, which are expensive in cost, labor and time, and often involve exposure to high radiation level (Hosono et al., 2021), aggravating the data scarcity problem.

Several neuroimaging initiatives such as ADNI (Mueller et al., 2005), OASIS (LaMontagne et al., 2019) and WRAP (Johnson et al., 2018) try to tackle this problem by collecting data from a large cohort over a decade with several follow-up visits from individual subjects. They consist of a population of sequences of samples that provides information on *longitudinal* disease trajectories. Previous longitudinal studies have yielded successful results such as predicting the future evolution of individuals at risk of AD (Marinescu et al., 2018) and imputing missing outcomes within trajectories (Luo et al., 2016) by fully exploiting the dataset when sufficient sample size became available.

Alternatively, rather than waiting for individuals to be affected by neurodegeneration, one may utilize generative methods to obtain real-like synthetic longitudinal samples. Conditional generative models (Hwang et al., 2019; Yang et al., 2021) have shown promising results in creating sequential data from scratch using time-varying diagnostic labels as conditions. However, these methods overlook the prior information of the condition which can be critical such as the irreversible nature of neurodegeneration. Moreover, these methods do not consider discontinuities in sample points caused by long intervals between clinical assessments (often ranging from 6 months to 6 years) (Mueller et al., 2005), and this inconsistency is a major practical issue in longitudinal disease analyses.

**Proposed method.** To address the issues discussed above, we propose **Con**ditional **D**iffusion model using **O**rdinal **R**egression (ConDOR), which generates long-term sequences conditioned on various ordinal factors. In this work, ConDOR is specifically applied to model the progression of degenerative disorders by synthesizing brain regional measurements based on disease-relevant metadata, such as age and sequential diagnostic labels that describe disease progression. These factors, which change monotonically over time (e.g., increasing age and deteriorating labels), serve as conditions to characterize disease trajectories and generate realistic brain measurements from imaging scans.

Specifically, ConDOR operates as an autoregressive model, sequentially generating samples from baseline to follow-up time points with a diffusion model to address the temporal heterogeneity among subjects. To tackle data discontinuities caused by long intervals between collections, ConDOR gradually generates unobserved interim samples, conditioned on interpolated ages and labels between consecutively observed time points. These pseudo-samples are crafted as a combination of two components: *cohort-level* and *subject-level* samples. A cohort-level sample is modeled with an ordinal regression (Winship & Mare, 1984) fitted to the entire population to explicitly learn the globally common transition pattern in ordinal factors. On the other hand, individual variability is incorporated as a subject-level sample obtained by interpolating samples in consecutive visits, which captures characteristics unique to individuals with aging and various disease manifestation patterns. Ultimately, these cohort and subject-level samples are combined to form unobserved interim pseudo-samples, and ConDOR iteratively estimates the difference in these consecutive pseudo-samples to reconstruct the subsequent observation at the next time point. Such concept of separately modeling *consistent* group-level and *variational* sample-level features has been traditionally studied in machine learning and computer vision, e.g., face recognition models that integrate common face with personal traits (Becerra-Riera et al., 2019; Yan et al., 2022) and graph models that aggregate features from global semantic clusters and local node similarities (Xu et al., 2021; Kuang et al., 2019).

Furthermore, to address the limited sample-size issues in medical data, we introduce an extended framework of ConDOR that learns data from multiple datasets by using an additional domain condition. This approach maximizes the utility of available data from diverse sources, thus enhancing the model's robustness and applicability. By flexibly utilizing both time-dependent (i.e., age and label) and time-independent conditions (i.e., domain), ConDOR can be potentially applied not only to various medical data but also to other data types where data are scattered over various sites, making it a versatile model for a wide range of applications.

**Contributions.** Our contributions are summarized as follows. **1)** We propose a novel generative model for longitudinal data, conditioned on ordinal factors such as age and disease severity. By incorporating ordinal regression into a diffusion model, our method effectively captures the ordinality of conditions. **2)** To handle temporal heterogeneity within visits, our method sequentially generates samples that seamlessly fill gaps within sparse data points. These interim samples capture both population-level trends and individual-specific features by a dual-sampling approach, which leads to generating personalized longitudinal data. **3)** To maximize the utility of limited data, the framework is extended to enable the model to learn data from different sources with a domain condition. By integrating both time-invariant and time-dependent factors, ConDOR improves generalizability across diverse datasets while capturing common progressive features.

As a result, ConDOR demonstrates its generality and efficacy on four independent AD biomarkers, provided by Alzheimer's Disease Neuroimaging Initiative (ADNI) and Open Access Series of Imaging Studies (OASIS). Also, ConDOR yields interpretable results at the brain regional level, validating that the generated results characterize realistic AD progression.

## 2   METHOD

In this section, we introduce "ConDOR", a method for generating sample sequences with time-varying ordinal conditions. We first introduce the details of the task that we aim to solve and the way of sampling from a conditional density function using Bayes' Theorem and ordinal regression, which forms the cornerstones of our model. Subsequently, we explain how the sampling strategy is integrated into a diffusion model, and further describe an extended version for learning data from multiple sources.

### 2.1   PROBLEM DEFINITION: DATA GENERATION WITH ORDINAL CONDITIONS

Consider a longitudinal sequence of samples $\{\boldsymbol{x}_t\}_{t=1}^{T}$, where the samples for $t = 1, ..., T$ time points ($T \geq 2$) are presented in chronological order. Each sample $\boldsymbol{x}_t \in \mathbb{R}^B$ represents a set of

independent features at time $t$, e.g., a set of brain measurements obtained from $B$ brain regions of interest (ROIs). For each time point, the samples come with ordinal conditions, e.g., age $\{a_t\}_{t=1}^{T}$ and diagnostic labels $\{y_t\}_{t=1}^{T}$ within a disease spectrum, where $y_t \in \{1, ..., C\}$ denotes the disease progression from the healthy control state ($y_t = 1$) to the most deteriorated state ($y_t = C$).

This study aims to characterize the progressive disease patterns and generate longitudinal sequences that comply with the ordinal conditions. Assuming that the brain regional measurements $\boldsymbol{x}_t$ as realizations of a random vector $\boldsymbol{X} = [X_1, ..., X_B]$ and age $a_t$ and disease stage $y_t$ are realizations of random variables $A$ and $Y$ respectively, a sample distribution is defined as $\boldsymbol{x}_t \sim f_{\boldsymbol{X}|A,Y}(\boldsymbol{x}_t|a_t, y_t)$. Here, the $f_{\boldsymbol{X}|A,Y}(\boldsymbol{x}_t|a_t, y_t)$ is a conditional probability density function (PDF) indicating the distribution of brain measurements for given age and disease stage. This formulation allows us to simulate realistic progression scenarios by considering both age and disease-specific variations in the brain.

## 2.2 Refining Conditional PDF with Bayes' Theorem

Using Bayes' Theorem (Bayes, 1763), the conditional PDF $f_{\boldsymbol{X}|A,Y}(\boldsymbol{x}_t|a_t, y_t)$ is formulated as

$$f_{\boldsymbol{X}|A,Y}(\boldsymbol{x}_t|a_t, y_t) = \frac{p_{Y|\boldsymbol{X},A}(y_t|\boldsymbol{x}_t, a_t) \cdot f_{\boldsymbol{X}|A}(\boldsymbol{x}_t|a_t)}{p_{Y|A}(y_t|a_t)}, \qquad (1)$$

which enables the decomposition of the complex, high-dimensional distribution into interpretable components. The first term $p_{Y|\boldsymbol{X},A}(y_t|\boldsymbol{x}_t, a_t)$ represents a probability mass function (PMF) that a subject is at a disease stage $y_t$ for given brain measurements and age. Given that $y_t$ is a discrete and ordinal variable, the PMF can be effectively modeled using an ordinal regression (Winship & Mare, 1984) to capture its ordinal nature. The second term $f_{\boldsymbol{X}|A}(\boldsymbol{x}_t|a_t)$ is a prior distribution characterizing age-relevant effects in estimating $\boldsymbol{x}_t$. As we do not assume any shape such as ordinality for $\boldsymbol{x}_t$, it needs to be directly estimated from given data using a non-parametric method such as kernel density estimation (KDE) (Parzen, 1962). Lastly, the denominator $p_{Y|A}(y_t|a_t) \triangleq \Pr(Y = y_t|A = a_t)$ represents the probability of being at stage $y_t$ given age $a_t$, which serves as a scaling constant. In the following sections, we introduce detailed methods of modeling $p_{Y|\boldsymbol{X},A}(y_t|\boldsymbol{x}_t, a_t)$ and $f_{\boldsymbol{X}|A}(\boldsymbol{x}_t|a_t)$ to estimate the conditional PDF $f_{\boldsymbol{X}|A,Y}(\boldsymbol{x}_t|a_t, y_t)$ and describe the sampling process of $\boldsymbol{x}_t$ from the estimated PDF using inverse transform sampling (Devroye, 1986).

**Modeling $p_{Y|\boldsymbol{X},A}(y_t|\boldsymbol{x}_t, a_t)$ with ordinal regression.** As $Y$ denotes discrete ordinal categories that describe the natural degenerative progression of a disease (i.e., healthy to disease), the distribution $p_{Y|\boldsymbol{X},A}(y_t|\boldsymbol{x}_t, a_t) \triangleq \Pr(Y = y_t|\boldsymbol{X} = \boldsymbol{x}_t, A = a_t)$ can be effectively estimated using an ordinal regression model. Unlike nominal classifiers that treat all labels independently, the ordinal regression accounts for the ordered nature of ranked groups, ensuring that gradual transitions in disease stages are captured. This approach leads to more accurate predictions and meaningful interpretations, as the model explicitly considers that the difference between non-adjacent stages (e.g., healthy vs. disease) is more significant than the difference between adjacent stages (e.g., healthy vs. prodromal). Also, since brain measurements $\boldsymbol{X}$ are key biomarkers that reflect the disease progression (Counts et al., 2017) and age $A$ is a risk factor of neurodegenerative diseases (Brown et al., 2005), using such factors as features of an ordinal regression enables effective estimation of the disease stage $Y$.

Given the ordered categories $y_t \in \{1, ..., C\}$, the probability of being at or below a certain stage $y_t$ is computed using cumulative probabilities as follows

$$\Pr(Y \leq y_t \,|\, \boldsymbol{X} = \boldsymbol{x}_t, A = a_t) = \Phi(\tau_{y_t} - \beta_{\boldsymbol{X}}^T \boldsymbol{x}_t - \beta_A^T a_t), \qquad (2)$$

where $\Phi(\cdot)$ is the logistic function defined as $\Phi(x) = \frac{1}{1+exp(-x)}$ and $\beta$'s are regression coefficients. The parameter $\tau_{y_t}$ is a threshold that separates the $y_t$-th stage from the next $(y_t + 1)$-th stage, satisfying $\tau_1 < \tau_2 < ... < \tau_{C-1}$. The above equation further allows us to derive the probability of being at the stage $y_t$ by taking the difference between consecutive cumulative probabilities as

$$\Pr(Y = y_t \,|\, \boldsymbol{X} = \boldsymbol{x}_t, A = a_t) = \Phi(\tau_{y_t} - \beta_{\boldsymbol{X}}^T \boldsymbol{x}_t - \beta_A^T a_t) - \Phi(\tau_{y_t-1} - \beta_{\boldsymbol{X}}^T \boldsymbol{x}_t - \beta_A^T a_t), \quad (3)$$

where $\tau_0 = -\infty$ and $\tau_C = \infty$. To obtain the optimal thresholds, this ordinal regression model is fitted for $N$ sequences with the following objective function

$$L(\theta, \beta_{\boldsymbol{X}}, \beta_A) = -\frac{1}{NT} \sum_{n=1}^{N} \sum_{t=1}^{T} \sum_{c=1}^{C} \mathbb{I}((y_t)^n = c) \, \log(\Pr(Y = c \,|\, \boldsymbol{X} = (\boldsymbol{x}_t)^n, A = (a_t)^n)), \quad (4)$$

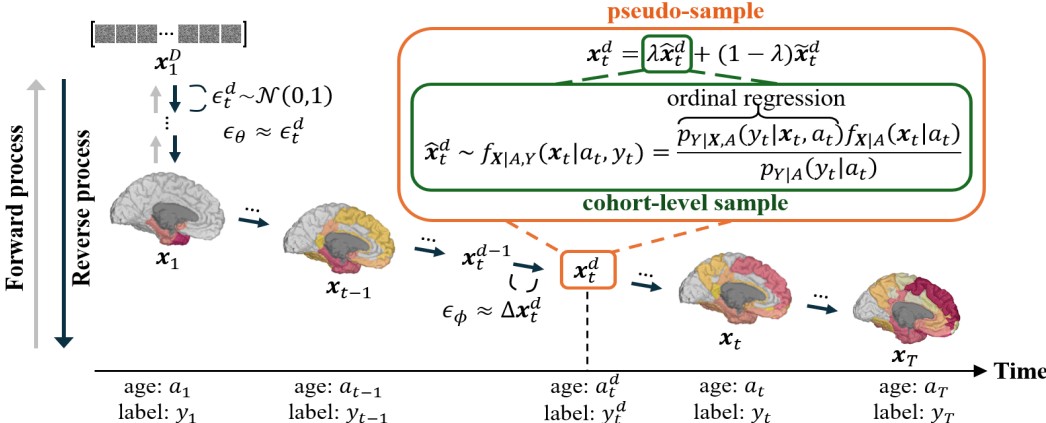

Figure 1: Overview of ConDOR. ConDOR generates longitudinal sequence $\{x_t\}_{t=1}^T$ conditioned on ordinal factors such as age $\{a_t\}_{t=1}^T$ and progressive labels $\{y_t\}_{t=1}^T$. Baseline data $x_1$ is generated by Regional Diffusion Model $\epsilon_\theta(\cdot)$ that estimates Gaussian noise $\epsilon_t^d \sim N(0,1)$. To capture temporal changes within a sequence, $x_t^d$ is sampled for every $d = 1, ..., D$ diffusion step between two data points adjacent in time $\{x_{t-1}, x_t\}$. The $x_t^d$ is a combination of cohort-level and subject-level samples (i.e., $\hat{x}_t^d$ and $\tilde{x}_t^d$) that considers general trend at the cohort-level and subject-specific progression for given conditions, respectively. Given the $x_t^d$, Temporal Diffusion Model $\epsilon_\phi(\cdot)$ learns the difference $\Delta x_t^d = x_t^d - x_t^{d-1}$ to generate follow-up data $\{x_t\}_{t=2}^T$.

where $\mathbb{I}(y_t = c)$ is an indicator function that outputs 1 if the label $y_t$ is $c$ and 0 otherwise. With this loss, thresholds are optimized to maximize the likelihood of accurately predicting ordinal samples.

**Modeling $f_{\boldsymbol{X}|A}(\boldsymbol{x}_t|a_t)$ via kernel density estimation.** As the distribution of $\boldsymbol{x}_t$ is unknown, we use a non-parametric KDE to estimate $f_{\boldsymbol{X}|A}(\boldsymbol{x}_t|a_t)$ from the available data. Given ROI-wise measurements $\boldsymbol{x}_t = [x_{t,1}, ..., x_{t,B}] \in \mathbb{R}^B$, the prior distribution $f_{\boldsymbol{X}|A}(\boldsymbol{x}_t|a_t)$ is derived as

$$f_{\boldsymbol{X}|A}(\boldsymbol{x}_t|a_t) = \prod_{b=1}^B f_{X_b|A}(x_{t,b}|a_t) = \prod_{b=1}^B \frac{1}{N} \sum_{n=1}^N \sum_{t=1}^T \frac{1}{h_b} k\left(\frac{x_{t,b} - (x_{t,b})^n}{h_b}\right), \qquad (5)$$

where $h_b$ is a bandwidth of the Gaussian kernel $k(\cdot)$. By integrating this distribution with the $p_{Y|\boldsymbol{X},A}(y_t|\boldsymbol{x}_t, a_t)$ from equation 3, the $f_{\boldsymbol{X}|A,Y}(\boldsymbol{x}_t|a_t, y_t)$ is derived as in equation 1.

**Inverse transform sampling of $\boldsymbol{x}$.** After estimating the conditional PDF $f_{\boldsymbol{X}|A,Y}(\boldsymbol{x}_t|a_t, y_t)$, inverse transform sampling (Devroye, 1986) is performed to generate the brain measurements $\boldsymbol{x}_t$ given its age and disease label. This is done by calculating its cumulative distribution function (CDF) as

$$F_{\boldsymbol{X}|A,Y}(\boldsymbol{x}_t|a_t, y_t) = \prod_{b=1}^B F_{X_b|A,Y}(x_{t,b}|a_t, y_t) = \prod_{b=1}^B \int_{-\infty}^{x_{t,b}} f_{X_b|A,Y}(u|a_t, y_t)du, \qquad (6)$$

which denotes the probability that $X_b$ take values less than or equal to $x_{t,b}$ for given conditions $a_t$ and $y_t$. Thus, given $\boldsymbol{u} = [u_1, ..., u_B] \sim \text{Uniform}(0,1)^B$, the inverse CDF $F_{\boldsymbol{X}|A,Y}^{-1}(\boldsymbol{u}|a_t, y_t)$ finds the $\boldsymbol{x}_t = [x_{t,1}, ..., x_{t,B}]$ such that each element satisfies $F_{X_b|A,Y}(x_{t,b}|a_t, y_t) = u_b$.

## 2.3 ConDOR: Conditional Diffusion model using Ordinal Regression

Fig. 1 illustrates the overall architecture of ConDOR that generates longitudinal samples $\{x_t\}_{t=1}^T$ given a sequence of ages $\{a_t\}_{t=1}^T$ and labels $\{y_t\}_{t=1}^T$. As the $\{x_t\}_{t=1}^T$ is an entangled representation of *both spatial and temporal features*, our method characterizes each feature by using two separate diffusion models: Regional Diffusion Model (RDM) $\epsilon_\theta(\cdot)$ and Temporal Diffusion Model (TDM) $\epsilon_\phi(\cdot)$, parameterized by $\theta$ and $\phi$, respectively. Specifically, ConDOR operates as an autoregressive model, where RDM generates the sample for baseline time point $x_1$, and TDM sequentially generates subsequent samples $\{x_t\}_{t=2}^T$. While RDM captures features from distinct regions within the data, TDM captures temporal features at both cohort-level and subject-level by combining two separate samples: one is derived from a whole cohort distribution and the other is estimated from an individual sequence. The details of each diffusion model are given below.

### 2.3.1 REGIONAL DIFFUSION MODEL FOR BASELINE DATA GENERATION

Since sample $\boldsymbol{x}_1$ at baseline time point is inherently absent of temporal prior features, RDM generates the baseline data by only accounting for their spatial representations. Building upon the Denoising Diffusion Probabilistic Model (DDPM) (Ho et al., 2020), RDM is crafted as a conditional diffusion model that treats $\boldsymbol{x}_t$ ($t = 1, ..., T$) as independent cross-sectional data. Its forward process $q(\cdot)$ gradually adds Gaussian noise $\epsilon_t^d \sim \mathcal{N}(0, 1)$ to $\boldsymbol{x}_t$ for $D$ diffusion steps as

$$q(\boldsymbol{x}_t^{1:D}|\boldsymbol{x}_t^0) := \prod_{d=1}^{D} q(\boldsymbol{x}_t^d|\boldsymbol{x}_t^{d-1}) := \prod_{d=1}^{D} \mathcal{N}(\boldsymbol{x}_t^d; \sqrt{1 - \beta_t^d}\boldsymbol{x}_t^{d-1}, \beta_t^d\mathbf{I}), \tag{7}$$

where $\boldsymbol{x}_t^0 = \boldsymbol{x}_t$ and $\beta_t^d$ is a variance schedule of the Gaussian distribution. During training, a reverse process $p_\theta(\cdot)$ reconstructs samples conditioned on $a_t$ and $y_t$ by estimating the noise as follows:

$$p_\theta(\boldsymbol{x}_t^{0:D}|a_t, y_t) := p(\boldsymbol{x}_t^D) \prod_{d=1}^{D} \mathcal{N}(\boldsymbol{x}_t^{d-1}; \mu_\theta(\boldsymbol{x}_t^d, a_t, y_t, d), \beta_t^d\mathbf{I}), \tag{8}$$

where the mean of Gaussian distribution $\mu_\theta(\boldsymbol{x}_t^d, a_t, y_t, d)$ is determined by a neural network $\epsilon_\theta(\boldsymbol{x}_t^d, a_t, y_t, d)$ (i.e., conditional U-Net (Ronneberger et al., 2015)). To reconstruct samples from a complete Gaussian noise, RDM is trained to estimate the diffusion step-wise noise $\epsilon_t^d$ with the following loss

$$L_{\text{RDM}} = \mathbb{E}_{\boldsymbol{x}_t, d, \epsilon_t^d \sim N(0,1)}||\epsilon_t^d - \epsilon_\theta(\boldsymbol{x}_t^d, a_t, y_t, d)||^2, \tag{9}$$

so that the baseline time point data are produced during the sampling process.

### 2.3.2 TEMPORAL DIFFUSION MODEL FOR FOLLOW-UP DATA GENERATION

To characterize temporal features within sequences, TDM is trained to learn changes among samples at different time points. As prior samples are highly likely to affect subsequent data, TDM operates as an autoregressive model that sequentially reconstructs the follow-up sample from the previous one, using the prior as a reference. Specifically, while RDM generates $\boldsymbol{x}_1$ from Gaussian noise, TDM generates $\boldsymbol{x}_t$ ($t \geq 2$) based on the prior data $\boldsymbol{x}_{t-1}$. In other words, TDM treats the previous data like a complete noise $\boldsymbol{x}_t^D$ in RDM, thereby eliminating the need for forward diffusion since the noise (i.e., baseline data) is already given. The reconstruction of follow-up data is done by a reverse diffusion for $D$ steps between every data pair adjacent in time $\{\boldsymbol{x}_{t-1}, \boldsymbol{x}_t\}_{t=2}^T$, i.e., sampling pseudo-samples $\{\boldsymbol{x}_t^d\}_{d=1}^D$ for each pair. Similar to RDM which estimates stepwise noises, TDM estimates the difference between $\boldsymbol{x}_t^{d-1}$ and $\boldsymbol{x}_t^d$, enabling the model to learn temporal changes efficiently.

Given that the diffusion steps $d = 1, ..., D$ of TDM correspond to the subdivided time between the real time points $t - 1$ and $t$, the diffusion step-wise age $a_t^d$ and label $y_t^d$ are defined as follows:

$$a_t^d = a_{t-1} + (a_t - a_{t-1})\frac{d}{D} \quad \text{and} \quad y_t^d = \min\left\{c \in \{1, ..., C\} : c \geq y_{t-1} + (y_t - y_{t-1})\frac{d}{D}\right\}, \tag{10}$$

The $a_t^d$ represents an interpolated age between $a_{t-1}$ and $a_t$, and the $y_t^d$ indicates a discretized label for the intermediate diffusion step. For example, if $y_{t-1}$ is 1 (e.g., healthy) and the following label $y_t$ is $C$ (e.g., the most deteriorated stage), $y_t^d$ sequentially advances from 1 to $C$ as the diffusion progresses. These interpolated conditions allow TDM to account for smooth transitions between different time points and incorporate temporal dynamics into the diffusion process.

**Integrating cohort-level sample & subject-level sample**. Given conditions $a_t^d$ and $y_t^d$, the pseudo-sample $\boldsymbol{x}_t^d$ at diffusion step $d$ is derived as a linear combination of $\hat{\boldsymbol{x}}_t^d$ and $\tilde{\boldsymbol{x}}_t^d$ as follows:

$$\boldsymbol{x}_t^d = \lambda\hat{\boldsymbol{x}}_t^d + (1 - \lambda)\tilde{\boldsymbol{x}}_t^d, \tag{11}$$

where $\hat{\boldsymbol{x}}_t^d \sim f_{\boldsymbol{X}|A,Y}(\boldsymbol{x}_t^d|a_t^d, y_t^d)$ is a *cohort-level* sample drawn from the overall cohort distribution defined in equation 1. This represents a sample that captures general trends of a population based on the given age and label. In contrast, $\tilde{\boldsymbol{x}}_t^d = \boldsymbol{x}_{t-1} + (\boldsymbol{x}_t - \boldsymbol{x}_{t-1})\frac{d}{D}$ is a *subject-level* sample, which is a set of interpolated brain measurements between two consecutive time points. This accounts for individual-specific changes in the brain over time. The $\lambda \in [0, 1]$ is a hyperparameter that balances the trade-off between cohort and subject-level information. Higher $\lambda$ emphasizes the global pattern at the cohort level, while lower $\lambda$ focuses more on the subject-specific trajectory. This approach enables the model to integrate both population-wide and individual-specific features flexibly.

Given the integrated sample $\boldsymbol{x}_t^d$, TDM estimates the differences $\Delta \boldsymbol{x}_t^d = \boldsymbol{x}_t^d - \boldsymbol{x}_t^{d-1}$ at each diffusion step $d = 1, .., D$, where $\boldsymbol{x}_t^0 = \boldsymbol{x}_{t-1}$ and $\boldsymbol{x}_t^D = \boldsymbol{x}_t$. For $t = 2, ..., T$, the model $\epsilon_\phi(\cdot)$ approximates the difference $\Delta \boldsymbol{x}_t^d$ with the following loss function

$$L_{\text{TDM}} = \mathbb{E}_{\boldsymbol{x}_t, d, \Delta \boldsymbol{x}_t^d} \left[ ||\Delta \boldsymbol{x}_t^d - \epsilon_\phi(\boldsymbol{x}_t^d, a_t, y_t, d)||^2 \right], \tag{12}$$

so that follow-up samples $\{\boldsymbol{x}_t\}_{t=2}^T$ are reconstructed by learning detailed changes within sequences.

## 2.4 MULTI-DOMAIN LEARNING WITH A DOMAIN CONDITION

Given that medical data analysis is inherently challenged by class imbalance and limited data availability due to costly data acquisition processes, integrating multi-source data is necessary to ensure robust and generalizable learning. Therefore, ConDOR provides an extended generative scheme by integrating longitudinal data from multiple domains while preserving domain-specific features. This is done by using an additional domain condition for reverse processes in Regional and Temporal Diffusion Models. Unlike time-varying age and label conditions, the domain does not depend on time, thereby an identical domain condition is consistently applied across all time points.

**Regional Diffusion Model with a domain condition**. Given $K$ domains, let $m_k$ $(k = 1, ..., K)$ be a domain type. The reverse process of RDM with this additional domain condition is defined as

$$p_\theta(\boldsymbol{x}_t^{0:D}|a_t, y_t, m_k) := p(\boldsymbol{x}_t^D) \prod_{d=1}^D \mathcal{N}(\boldsymbol{x}_t^{d-1}; \mu_\theta(\boldsymbol{x}_t^d, a_t, y_t, m_k, d), \beta_t^d \mathbf{I}), \tag{13}$$

where the $\mu_\theta$ is determined by a neural network $\epsilon_\theta(\cdot)$ trained with the following denoising objective

$$L_{\text{RDM}} = \mathbb{E}_{\boldsymbol{x}_t, d, \epsilon_t^d \sim N(0,1)} ||\epsilon_t^d - \epsilon_\theta(\boldsymbol{x}_t^d, a_t, y_t, m_k, d)||^2. \tag{14}$$

This allows RDM to estimate the noise $\epsilon_t^d$ while incorporating the domain condition, so that RDM generates domain-specific data, enhancing the quality and relevance of the generated samples.

**Temporal Diffusion Model with a domain condition**. Let $N_k$ be the number of sequences for domain $m_k$, and $M$ be a random variable that denotes the domain condition. With the $M$, the equation 1 is rewritten as a mixture distribution $\bar{f}_{\boldsymbol{X}|A,Y,M}(\cdot)$ as follows:

$$\bar{f}_{\boldsymbol{X}|A,Y,M}(\boldsymbol{x}_t|a_t, y_t, m_k) = \sum_{k=1}^K \alpha_k \, f_{\boldsymbol{X}|A,Y,M}^k(\boldsymbol{x}_t|a_t, y_t, m_k), \tag{15}$$

where $\alpha_k = \frac{N_k}{\sum_{k=1}^K N_k}$ is a mixing coefficient for a domain-specific PDF $f_{\boldsymbol{X}|A,Y,M}^k(\cdot)$. This mixture density allows small sample-sized domains (or domains with class imbalances) to be enriched by incorporating other larger domains (or domains with more balanced classes), thereby mitigating the risk of overfitting to limited data. From the mixture density, a cohort-level sample $\hat{\boldsymbol{x}}_t^d \sim \bar{f}_{\boldsymbol{X}|A,Y,M}(\boldsymbol{x}_t^d|a_t^d, y_t^d, m_k)$ is obtained to further construct the $\boldsymbol{x}_t^d$ as in equation 11. Along with the age and label conditions, the domain condition $m_k$ is inputted into TDM $\epsilon_\phi(\cdot)$ to estimate the diffusion step-wise difference $\Delta \boldsymbol{x}_t^d = \boldsymbol{x}_t^d - \boldsymbol{x}_t^{d-1}$ by using the following loss function

$$L_{\text{TDM}} = \mathbb{E}_{\boldsymbol{x}_t, d, \Delta \boldsymbol{x}_t^d} ||\Delta \boldsymbol{x}_t^d - \epsilon_\phi(\boldsymbol{x}_t^d, a_t, y_t, m_k, d)||^2. \tag{16}$$

With this loss, the generated samples accurately reflect the underlying characteristics of each domain while still accounting for the ordinal nature of the age and label conditions.

## 3 RELATED WORKS

**Longitudinal Data Analysis.** In contrast to cross-sectional studies that focus on a snapshot of data at a single point in time, longitudinal data analyses focus on understanding temporal dynamics within data over time. This approach is widely used in modeling time-series data in various domains such as healthcare (Mosquera et al., 2023; Yoon et al., 2023; Hwang et al., 2019), traffic flow forecasting (Shu et al., 2021; Li & Zhu, 2021), and computer vision tasks such as human motion recognition in videos (Chen et al., 2021; Cheng et al., 2020). In particular, studies in the healthcare domain include tracking electronic health records (Joshua Lin et al., 2022), predicting adolescent brain development (Holm et al., 2023), and modeling the progression of diseases (Hwang et al.,

2019). By tracking health outcomes over time, these long-term studies contribute to the understanding of the progression of disorders and provide critical insights into how patient outcomes evolve.

**Generative Models for Tabular Data.** Tabular data are ubiquitous in various fields such as medicine, economics, and marketing, often involving a mix of discrete and continuous variables. Recent generative methods handle both types simultaneously (Xu et al., 2019; Zhao et al., 2021; Kotelnikov et al., 2023), rather than focusing on either type (Choi et al., 2017). While these methods have shown successful results on tabular data, many are limited to cross-sectional data, failing to capture temporal dynamics in longitudinal data. This shortfall is particularly critical in healthcare, where understanding the evolution of patient health is essential for precise prognosis and treatment planning. To address the challenge, our method focuses on generating realistic longitudinal tabular data (i.e., biomarkers) that not only considers the intricate relationships between heterogeneous variables but also accurately characterizes temporal features across different time points.

## 4 EXPERIMENT

In this section, we describe the quantitative comparison of ConDOR with nine baseline methods and discuss the effect of model components along with interpretations. Overall, we performed experiments on four different biomarkers from MRI and PET images provided by two independent longitudinal Alzheimer's disease (AD) studies: Alzheimer's Disease Neuroimaging Initiative (ADNI) and Open Access Series of Imaging Studies (OASIS), whose demographics and preprocessing methods are reported in Appendix A. Also, we provide detailed implementation settings of ConDOR in Appendix B and extensive qualitative comparisons with the baseline models in Appendix. D.

### 4.1 DATASETS

**ADNI.** ADNI study (Mueller et al., 2005) is the largest long-term study of AD, aimed at collecting a comprehensive set of biomarker data from participants over an extended period. Four AD-associated biomarkers (Ortner et al., 2019) collected from MRI and PET scans were used: (1) Cortical Thickness (CT) from MRI, Standardized Uptake Value Ratio (SUVR) of (2) Amyloid, (3) fluorodeoxyglucose (FDG), and (4) Tau from PET. From $N$=2153 subjects, we excluded those with a single time point, resulting in 178, 687, 678, and 166 subjects for CT, Amyloid, FDG, and Tau respectively, with time points $T$ per participant ranging from 2 to 10. All biomarkers were measured from 148 ROIs parcellated based on the Destrieux atlas (Destrieux et al., 2010). Five diagnostic labels were used: Cognitively Normal (CN), Significant Memory Concern (SMC), Early Mild Cognitive Impairment (EMCI), Late MCI (LMCI), and AD, with the disease progressing irreversibly from CN to AD.

**OASIS.** Compared to the ADNI, OASIS dataset (LaMontagne et al., 2019) provides a relatively small data where only Tau is available for $N$=32 subjects. All subjects have two time points, spaced 6 months apart, each labeled CN or AD. Among the 32 subjects, three were diagnosed with AD, where only one of them was consistently diagnosed with AD at both time points, while the other two transitioned from CN to AD. As in the ADNI, Tau was measured at 148 ROIs based on the Destrieux atlas. Due to the small sample size with a biased label distribution, single-domain learning on the OASIS dataset is highly prone to overfitting. Thus, we performed multi-domain learning on the Tau of ADNI and OASIS together with a domain condition to improve the model's generalizability.

### 4.2 EXPERIMENTAL SETUP

**Baseline methods.** We utilized nine baseline models, encompassing various types of generative models such as normalizing flow, generative adversarial networks (GANs), variational autoencoders (VAEs), diffusion models, and traditional interpolation-based synthetic techniques. Specifically, SMOTE (Chawla et al., 2002) is an interpolation-based method that synthesizes data points as a convex combination of a real data point with its $k$-th nearest neighbor. This is a simple yet effective solution as demonstrated in (Camino et al., 2020; Kotelnikov et al., 2023), where it outperforms GANs and diffusion models for tabular data generation. Also, we adopt Conditional Recurrent Flow (CRow) (Hwang et al., 2019), a conditional normalizing flow model that generates long-term sequences. We also included GANs such as Conditional Tabular GAN (CTGAN) (Xu et al., 2019), Conditional Table GAN (CTAB-GAN) (Zhao et al., 2021), and CTAB-GAN+ (Zhao et al., 2024). For VAEs, we adopt Generative Modelling with Graph Learning (GOGGLE) (Liu et al., 2023) and TVAE, which is VAE (Kingma, 2013) for tabular data generation introduced in (Xu et al., 2019). Lastly, diffusion-based models such as TabDDPM (Kotelnikov et al., 2023) and a conditional form of Denoising Diffusion Probabilistic Model (DDPM) (Ho et al., 2020) are adopted.

Table 1: Generation performance for single-domain learning on CT, Amyloid, and FDG test sets, with averages and standard deviations from three replicates. The best results are in bold, and the second-best are underlined.

| Model | Cortical Thickness | | | Amyloid | | | FDG | | |
|---|---|---|---|---|---|---|---|---|---|
| | WD | RMSE | JSD | WD | RMSE | JSD | WD | RMSE | JSD |
| SMOTE | $8.68 \pm 0.0$ | $0.57 \pm 0.0$ | $0.025 \pm 0.0$ | $5.82 \pm 0.0$ | $0.34 \pm 0.0$ | $0.014 \pm 0.0$ | $3.19 \pm 0.0$ | $0.10 \pm 0.0$ | $0.002 \pm 0.0$ |
| CRow | $5.13 \pm 0.13$ | $0.78 \pm 0.014$ | $0.042 \pm 0.001$ | $5.37 \pm 0.40$ | $0.85 \pm 0.023$ | $0.061 \pm 0.011$ | $2.86 \pm 0.04$ | $0.37 \pm 0.007$ | $0.012 \pm 0.0$ |
| CTGAN | $7.90 \pm 0.12$ | $0.37 \pm 0.005$ | $0.022 \pm 0.001$ | $11.00 \pm 0.02$ | $0.43 \pm 0.001$ | $0.036 \pm 0.0$ | $4.52 \pm 0.02$ | $0.13 \pm 0.001$ | $0.003 \pm 0.0$ |
| CTAB-GAN | $6.43 \pm 0.29$ | $0.35 \pm 0.033$ | $0.016 \pm 0.001$ | $7.93 \pm 0.63$ | $0.42 \pm 0.034$ | $0.023 \pm 0.004$ | $3.47 \pm 0.03$ | $0.11 \pm 0.002$ | $0.002 \pm 0.0$ |
| CTAB-GAN+ | $6.09 \pm 0.40$ | $0.34 \pm 0.054$ | $0.014 \pm 0.001$ | $6.50 \pm 0.29$ | $0.35 \pm 0.017$ | $0.015 \pm 0.001$ | $3.58 \pm 0.03$ | $0.11 \pm 0.007$ | $0.002 \pm 0.0$ |
| GOGGLE | $8.05 \pm 0.39$ | $0.34 \pm 0.016$ | $0.020 \pm 0.001$ | $16.62 \pm 0.16$ | $0.65 \pm 0.006$ | $0.010 \pm 0.0$ | $9.33 \pm 0.22$ | $0.25 \pm 0.006$ | $0.010 \pm 0.002$ |
| TVAE | $5.69 \pm 0.59$ | $0.38 \pm 0.053$ | $0.013 \pm 0.003$ | $7.24 \pm 0.21$ | $0.39 \pm 0.030$ | $0.015 \pm 0.002$ | $4.52 \pm 0.42$ | $0.16 \pm 0.019$ | $0.004 \pm 0.001$ |
| DDPM | $9.71 \pm 3.94$ | $0.59 \pm 0.160$ | $0.028 \pm 0.017$ | $9.34 \pm 2.51$ | $0.66 \pm 0.282$ | $0.026 \pm 0.015$ | $2.90 \pm 0.46$ | $0.10 \pm 0.015$ | $\mathbf{0.001 \pm 0.001}$ |
| TabDDPM | $14.23 \pm 0.02$ | $0.61 \pm 0.0$ | $0.070 \pm 0.001$ | $31.28 \pm 0.05$ | $1.18 \pm 0.001$ | $0.145 \pm 0.001$ | $11.59 \pm 0.04$ | $0.32 \pm 0.001$ | $0.018 \pm 0.0$ |
| ConDOR | $\mathbf{4.25 \pm 0.02}$ | $\mathbf{0.21 \pm 0.002}$ | $\mathbf{0.007 \pm 0.001}$ | $5.27 \pm 0.09$ | $\mathbf{0.31 \pm 0.012}$ | $\mathbf{0.008 \pm 0.001}$ | $\mathbf{2.49 \pm 0.02}$ | $\mathbf{0.09 \pm 0.001}$ | $0.001 \pm 0.0$ |

Table 2: Generation performance for multi-domain learning on Tau of ADNI and OASIS test sets. The average of three replicates and their standard deviations are reported along with the generation time for sampling 36 sequences (i.e., the number of test data) and the average time per sequence.

| Model | WD | RMSE | JSD | Test set gen. time (s) | Per-seq. gen. time (s) |
|---|---|---|---|---|---|
| SMOTE | $7.146 \pm 0.0$ | $0.460 \pm 0.0$ | $0.029 \pm 0.0$ | 29.85 | 0.83 |
| CRow | $9.494 \pm 0.188$ | $0.599 \pm 0.027$ | $0.045 \pm 0.045$ | $\mathbf{1.39}$ | $\mathbf{0.04}$ |
| CTGAN | $19.545 \pm 0.416$ | $0.441 \pm 0.014$ | $0.280 \pm 0.025$ | 2.97 | 0.08 |
| CTAB-GAN | $8.950 \pm 0.512$ | $0.455 \pm 0.036$ | $0.078 \pm 0.011$ | 26.60 | 0.74 |
| CTAB-GAN+ | $7.738 \pm 1.211$ | $0.393 \pm 0.048$ | $0.034 \pm 0.005$ | 48.90 | 1.36 |
| GOGGLE | $8.380 \pm 0.445$ | $0.343 \pm 0.014$ | $\mathbf{0.012 \pm 0.0}$ | $\underline{2.08}$ | $\underline{0.06}$ |
| TVAE | $\mathbf{5.096 \pm 0.303}$ | $\underline{0.300 \pm 0.020}$ | $\underline{0.018 \pm 0.003}$ | 4.33 | 0.12 |
| DDPM | $9.671 \pm 1.738$ | $0.649 \pm 0.235$ | $0.032 \pm 0.014$ | 81.12 | 2.25 |
| TabDDPM | $50.592 \pm 0.404$ | $1.890 \pm 0.018$ | $0.400 \pm 0.0$ | 151.09 | 4.20 |
| ConDOR (Ours) | $\underline{5.625 \pm 0.040}$ | $\mathbf{0.293 \pm 0.011}$ | $\mathbf{0.012 \pm 0.0}$ | 65.7 | 1.83 |

**Evaluation.** Following a prior work (Kotelnikov et al., 2023), three metrics were used to evaluate the difference between generated samples and the test data: (1) Wasserstein distance (**WD**), (2) Root mean squared error (**RMSE**), and (3) Jensen-Shannon divergence (**JSD**). WD measures the expected minimum distance to transform one distribution into another, primarily reflecting differences in the global structure of the two distributions. In contrast, RMSE emphasizes detailed sample-wise differences, as it calculates the squared error for each pair of points. Similarly, JSD is sensitive to imbalances between distributions, capturing the differences in distributional shape and entropy. For all experiments, we used 80% of the whole data for training and the rest 20% for testing. The number of sampled data is equal to the number of test data in each respective experiment. All baselines and our model were trained three times to report their average results with standard deviation.

## 4.3 QUANTITATIVE RESULTS

**Single-domain learning.** Table 1 presents the quantitative results from the single-domain experiments on CT, Amyloid, and FDG. In all experiments and across all metrics, ConDOR consistently outperformed all baseline methods. Specifically, in the CT experiment, ConDOR demonstrated far smaller RMSE by a margin of 0.57 over CRow and at least 0.13 over GOGGLE and CTAB-GAN+. In the Amyloid experiment, ConDOR achieved up to 26.01 lower WD than TabDDPM and at least 0.1 lower than CRow. Although the margin over CRow is small on WD, our method significantly outperformed CRow on RMSE and JSD by margins of 0.54 and 0.053, respectively. Note that RMSE and JSD are sensitive to sample-wise differences and distributional imbalances, while WD mainly focuses on global structural differences, such as mean differences between distributions. Thus, these results suggest that ConDOR is more effective in *capturing individual characteristics with high variance* compared to CRow. In the FDG experiment, ConDOR surpassed all baselines in WD and RMSE, and both our method and DDPM achieved the best result on JSD.

**Multi-domain learning.** As the sample size of the OASIS is small and label distribution is biased, the Tau of ADNI and OASIS were merged into a unified dataset and learned together using a domain condition. Table 2 shows quantitative results from the unified dataset and sampling times. ConDOR surpassed eight out of nine baselines across all distance metrics and achieved comparable results to GOGGLE and TVAE. Specifically, our method and GOGGLE showed the same 0.012 on JSD, and TVAE surpassed our method on WD by ∼0.55. However, ConDOR far outperformed GOGGLE on WD and RMSE by ∼2.7 and 0.05, respectively, and exceeded TVAE on RMSE and JSD. These results suggest that although TVAE better estimates the overall distribution shape, our method excels in preserving finer details by capturing outliers and subtle differences at specific data points.

To assess computational efficiency, we measured the time taken to generate 36 sequences, corresponding to the test data size. As a result, CRow was the fastest taking ∼1.4 seconds, and our

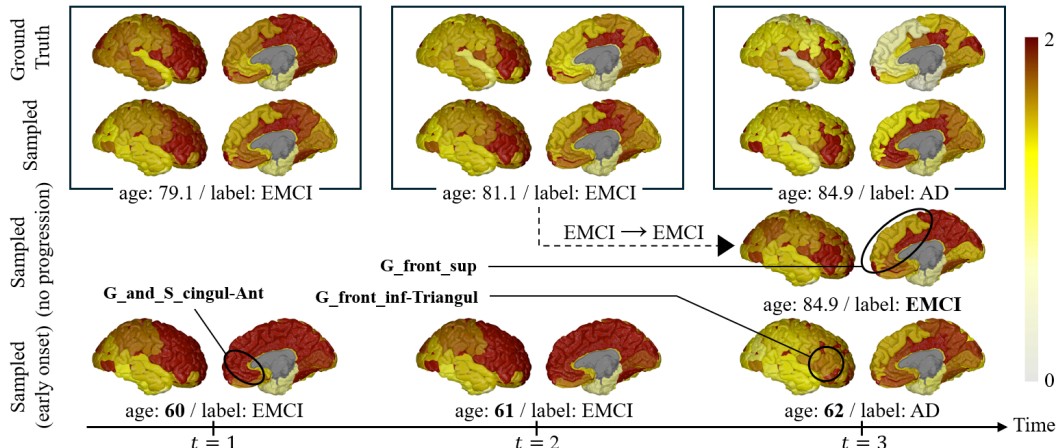

Figure 2: Visualization of generation results from the FDG experiment (subject ID: 130_S_2403). The first row shows the ground truth sequence with three time points and the second panel shows the generated sequence from ConDOR for the same temporal conditions which highly resemble the real samples. In the third row, brain measurements are generated assuming the disease has not manifested at $t = 3$, while the bottom shows the sampled results with the same label condition as the ground truth with younger ages in sixties.

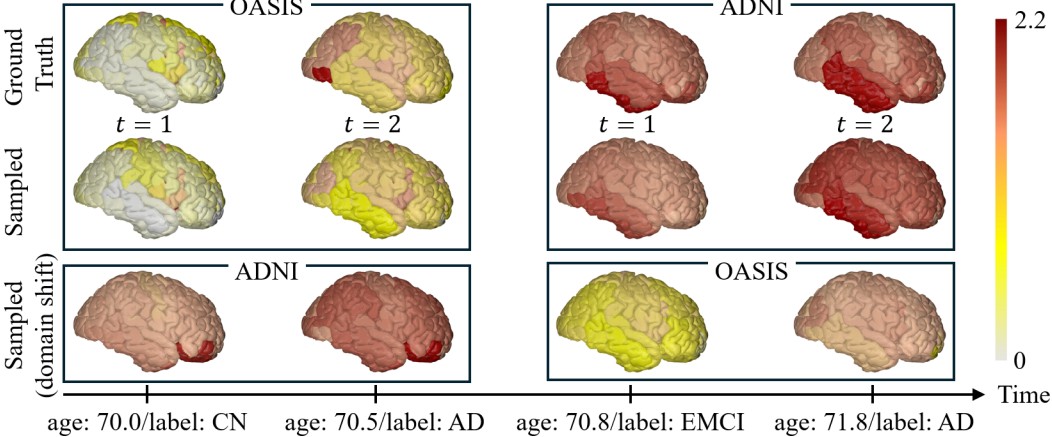

Figure 3: Visualization of generated results from the Tau experiment and results with switched domain conditions. The top panel is the ground truth of two samples with two time points (subject ID: OAS30818 and 129_S_4422 for OASIS and ADNI, respectively), where the data from different domains show different characteristics. The second panel shows generated results from ConDOR, and the bottom panel shows results with reversed domain conditions while age and label are consistent.

method took ∼66 seconds. In general, this increased sampling time was typical for diffusion-based methods such as DDPM, TabDDPM, and ConDOR, due to the iterative denoising steps in the sampling process. However, as shown in both Table 1 and Table 2, the sample quality of our method far surpasses other models, showing a trade-off between quality and computation speed.

### 4.4 MODEL BEHAVIOR AND ABLATION STUDY

**Analyses on conditions.** Figures 2 and 3 illustrate qualitative results from ConDOR on the FDG and Tau experiments, respectively, highlighting the impact of label, age, and domain conditions. Given that FDG SUVR generally decreases with AD progression (Mosconi et al., 2009) while Tau SUVR increases (Sjögren et al., 2001), these trends are accurately reflected in the generated results. For example, the top row in Fig. 2 is the ground truth and the second row is the generated sequence from ConDOR with the same conditions, where regional results resemble the ground truth for all time points. The third row illustrates a label-consistent scenario, where the disease has not progressed from EMCI to AD. In this case, the FDG SUVR at $t = 3$ remains higher than that of AD in the second panel, which demonstrates the sensitivity of ConDOR that captures subtle differences between labels. One of the most prominent differences is shown in *the superior frontal gyrus*, where significant glucose reduction has been observed along the AD progression in various AD studies (Sanabria-Diaz et al., 2013; He et al., 2015). While both the ground truth and generated

Table 3: Ablation studies on the cohort-level sample weight $\lambda$ for all experiments.

| $\lambda$ | ADNI | | | | | | ADNI+OASIS | |
|---|---|---|---|---|---|---|---|---|
| | Cortical Thickness | | Amyloid | | FDG | | Tau | |
| | WD | RMSE | WD | RMSE | WD | RMSE | WD | RMSE |
| 0 | $4.423 \pm 0.115$ | $0.208 \pm 0.003$ | $5.608 \pm 0.070$ | $0.321 \pm 0.010$ | $2.551 \pm 0.072$ | $\underline{0.086 \pm 0.002}$ | $6.628 \pm 0.286$ | $0.334 \pm 0.016$ |
| 0.1 | $\mathbf{4.252 \pm 0.024}$ | $\mathbf{0.205 \pm 0.002}$ | $\mathbf{5.400 \pm 0.139}$ | $0.313 \pm 0.012$ | $\mathbf{2.487 \pm 0.015}$ | $\mathbf{0.085 \pm 0.001}$ | $6.172 \pm 0.188$ | $0.320 \pm 0.007$ |
| 0.3 | $4.368 \pm 0.135$ | $0.208 \pm 0.007$ | $\underline{5.522 \pm 0.173}$ | $\mathbf{0.296 \pm 0.012}$ | $2.512 \pm 0.002$ | $0.086 \pm 0.002$ | $5.833 \pm 0.163$ | $0.313 \pm 0.007$ |
| 0.5 | $4.372 \pm 0.064$ | $\underline{0.207 \pm 0.004}$ | $5.541 \pm 0.088$ | $0.326 \pm 0.088$ | $2.595 \pm 0.031$ | $0.088 \pm 0.001$ | $\underline{5.692 \pm 0.069}$ | $\underline{0.305 \pm 0.008}$ |
| 0.7 | $4.496 \pm 0.035$ | $0.215 \pm 0.002$ | $5.530 \pm 0.086$ | $0.318 \pm 0.012$ | $2.567 \pm 0.024$ | $0.087 \pm 0.001$ | $\mathbf{5.645 \pm 0.024}$ | $\mathbf{0.292 \pm 0.010}$ |
| 0.9 | $4.372 \pm 0.083$ | $0.217 \pm 0.012$ | $5.629 \pm 0.029$ | $\underline{0.309 \pm 0.007}$ | $2.632 \pm 0.108$ | $0.088 \pm 0.002$ | $5.999 \pm 0.223$ | $0.313 \pm 0.003$ |
| 1 | $4.432 \pm 0.122$ | $0.209 \pm 0.005$ | $5.852 \pm 0.287$ | $0.331 \pm 0.008$ | $2.620 \pm 0.046$ | $0.088 \pm 0.002$ | $5.832 \pm 0.207$ | $0.313 \pm 0.023$ |

results with the AD label show relatively low FDG SUVR in this area, the sampled EMCI results at the same age display higher values, resembling the ground truth of the EMCI.

On the other hand, in the early-onset AD (EOAD) scenario (i.e., AD diagnosed before the age of 65) on the bottom row, ROIs such as *the cingulate gyrus and sulcus* show higher FDG SUVR than the ground truth, which is known to be a highly negatively correlated region with age (Jiang et al., 2018). Also, it is worth noting that *the triangular part of the inferior frontal gyrus* of the EOAD shows lower values than late-onset AD (LOAD). This result aligns with established AD studies (Kim et al., 2005; Kalpouzos et al., 2005), which have shown that EOAD features more focal reductions in glucose metabolism in *the frontal cortex*, while LOAD tends to have more diffuse hypometabolism.

Fig. 3 illustrates results from the Tau experiment involving two samples each from the OASIS and ADNI datasets. Both samples span two time points with similar ages, yet the OASIS sample shifts from CN to AD, while the ADNI sample progresses from EMCI to AD. In the top row, the ground truth for AD of OASIS (2nd column) and AD of ADNI (4th column) show significant differences, despite similar ages, possibly due to differences in the data collection environment.

As shown in the second row, our method successfully captured these domain-specific differences. Moreover, when domain conditions are reversed (3rd row), ConDOR not only characterizes the distinct features of each domain but also accurately captures the positive correlation between Tau SUVR and disease progression. Specifically, the bottom left sample with the ADNI and CN conditions displays lower Tau SUVR than the ground truth of the ADNI EMCI (top, 3rd column). Similarly, the EMCI sample of OASIS (bottom, 3rd column) shows intermediate values between the CN and AD of OASIS. Given that the original OASIS dataset has only CN and AD labels, multi-domain learning with other sources, such as ADNI–which includes intermediate MCI labels–allows us to generate a broader spectrum of data. Such an approach empowers the model to learn the comprehensive, full progression from CN to AD, thereby enhancing its generalizability across different disease stages.

**Effect of the cohort-level weight** $\lambda$. Table 3 presents the result of the ablation study on the $\lambda$, which balances the effect of cohort-level and subject-level samples used in TDM. For all experiments, using both samples (i.e., $0 < \lambda < 1$) outperformed the settings using only subject-level (i.e., $\lambda = 0$) or cohort-level samples (i.e., $\lambda = 1$) alone. In the single-domain experiments on CT, Amyloid, and FDG, $\lambda = 0.1$ yielded the best results. On the other hand, in the multi-domain experiment with Tau, higher $\lambda$'s such as $\lambda = 0.7$ and $0.5$ produced the best and second-best results, respectively. These findings suggest that using both cohort-level and individual-specific features improves the sample quality regardless of the domain settings. However, subject-specific details play a relatively critical role in single-domain learning, while capturing broader population-wide trends across multiple domains is more important when learning from diverse sources for effective generalization. Thus, by adjusting the $\lambda$, the model can generate higher-quality samples across both homogeneous and heterogeneous data, with the choice of $\lambda$ varying based on the task and data distribution.

## 5 CONCLUSION

In this paper, we introduced ConDOR, a novel conditional diffusion model for generating longitudinal samples conditioned on multiple ordinal factors. When applied to the generation of costly longitudinal neurodegenerative disease data, ConDOR addresses the data scarcity issue in medical data analyses. By incorporating an ordinal regression model into the diffusion process, ConDOR effectively characterizes the realistic ordinal dynamics of the disease. Also, the model generates smooth temporal samples by employing a dual-sampling strategy, which blends both individual-specific and global characteristics obtained from the entire sample distribution. As a result, ConDOR outperformed nine recent generative methods, underscoring its potential to enhance our understanding of the complex relationships between heterogeneous disease-related factors and disease progression.

**Reproduciblity Statement.** To ensure the reproducibility of our work, we provide a detailed dataset description in Appendix A, and implementation settings including all hyperparameters in Appendix B. Regarding quantitative evaluations, all experiments were replicated three times and their mean with standard deviation is reported. Pretrained ConDOR models for all datasets, and codes of ConDOR and baseline methods are availble at `https://github.com/Hannah37/ConDOR-ICLR25`.

**Acknowledgments.** This research was supported by NRF-2022R1A2C2092336 (70%), RS-2022-II2202290 (20%), and RS-2019-II191906 (AI Graduate Program at POSTECH, 10%).

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

In the appendix and supplementary material, we present **1)** the demographics of four biomarkers provided by ADNI and OASIS studies, **2)** detailed implementation settings of ConDOR used for each experiment, **3)** limitations of ConDOR and suggestions for future work, **4)** extensive qualitative comparisons with baseline methods, and **5)** a video illustrating a sampled sequence that shows changes in Amyloid SUVR over 20 years, which were not included in the main manuscript due to the page limit.

## A    DATASET DEMOGRAPHICS

ADNI (Mueller et al., 2005) is the largest open-source AD study, providing longitudinal and multisite neuroimaging data collected from 21 study sites. We utilized four AD-associated biomarkers from the ADNI dataset: (1) cortical thickness (CT) from MRI, (2) SUVR of Amyloid from PET, (3) SUVR of fluorodeoxyglucose (FDG) from PET, and (4) SUVR of Tau from PET. A total of 2153 subjects provided at least one MRI/PET image, with 1101 providing MRI for CT, 1205 providing Amyloid-PET, 1447 providing FDG-PET, and 549 providing Tau-PET. From the brain scans, we performed preprocessing to obtain brain regional measurements. We used the Destrieux atlas (Destrieux et al., 2010) to parcellate the cortical surface into 148 regions using T1-weighted MR images by skull stripping, tissue segmentation, and image registration. Based on the tissue segmentation result, we measured the average cortical thickness in each region by Freesurfer. For the Amyloid, FDG, and Tau, we calculated the region-wise average concentration level from PET scans. The cerebellum was used as the reference region to calculate the SUVR for each pathology modality.

Subsequently, we removed subjects whose age or label is missing and those who have only one time point. Consequently, we obtained 178, 687, 678, and 166 subject data for CT, Amyloid, FDG, and Tau respectively. The demographic details for these biomarkers are summarized in Table 4. The number of time points varies across subjects, with a minimum number of two for all biomarkers. The maximum number of time points is five for Amyloid, Tau, and CT, and ten for FDG. The average time interval between data collection points and their standard deviations are as follows: $2.2 \pm 0.78$ years for CT, $2.2 \pm 0.72$ years for Amyloid, $1.3 \pm 1.20$ years for FDG, and $1.2 \pm 0.37$ years for Tau.

The OASIS dataset (LaMontagne et al., 2019) provided Tau-PET scans from 81 subjects. To calculate regional Tau SUVR, we registered the PET scans to T1-weighted images and computed standard uptake values (SUV) for each region based on the Destrieux atlas. Similar to the ADNI preprocessing, we used the cerebellum as the reference region to compute the SUV ratio (SUVR). After excluding subjects whose age or label is missing and those with only a single time point, we retained 32 subjects. All subjects have two time points with a 6-month interval between data collection. Of the 32 subjects, three were diagnosed with AD; one had consistent AD diagnosis at both time points, while the other two transitioned from CN to AD. The demographic information for the OASIS dataset is presented in Table 5.

## B    DETAILED IMPLEMENTATION SETTING

To train ConDOR, we used PyTorch with a single NVIDIA GeForce RTX 3090 GPU. For all biomarkers, we applied a stratified train-test split to the data, maintaining the ratio of the given five labels, with 80% of data used for training and the remaining 20% for testing. Also, for the multi-domain learning on Tau, we first combined the Tau datasets from ADNI and OASIS, and then applied a stratified train-test split to the unified dataset maintaining the sample size ratio of each domain. To prevent biased results, all experiments were replicated three times with different parameter initialization, and their average performance along with standard deviation were reported. In Table 6, we provide details of the implementation settings of ConDOR for all experiments on four biomarkers. We performed a grid search to choose the best learning rate in $\{0.002, 0.0015, 0.001, 0.0008, 0.0005\}$, and a batch size in $\{16, 32\}$. In the reverse process of RDM and TDM, U-Net (Ronneberger et al., 2015) is used to implement $\epsilon_\theta(\cdot)$ and $\epsilon_\phi(\cdot)$.

Table 4: Demographics of the ADNI dataset. The numbers of subjects for each category were calculated based on the baseline time point samples. The number of records denotes the total number of cross-sectional samples across the population.

| Biomarker | Category | CN | SMC | EMCI | LMCI | AD |
|---|---|---|---|---|---|---|
| Cortical Thickness | # of subjects | 50 | 32 | 64 | 23 | 9 |
| | # of records | 112 | 72 | 152 | 56 | 39 |
| | Gender (M / F) | 26/24 | 24/8 | 40/24 | 13/10 | 5/4 |
| | Age (Mean $\pm$ Std) | $71.3 \pm 4.2$ | $72.1 \pm 4.7$ | $72.6 \pm 6.9$ | $74.2 \pm 5.2$ | $76.5 \pm 7.1$ |
| Amyloid | # of subjects | 150 | 87 | 238 | 158 | 54 |
| | # of records | 405 | 231 | 715 | 376 | 270 |
| | Gender (M / F) | 79/71 | 56/31 | 138/100 | 88/70 | 31/23 |
| | Age (Mean $\pm$ Std) | $71.9 \pm 3.9$ | $73.0 \pm 4.9$ | $73.3 \pm 7.1$ | $74.1 \pm 7.9$ | $76.0 \pm 7.5$ |
| FDG | # of subjects | 147 | 4 | 151 | 262 | 114 |
| | # of records | 415 | 4 | 345 | 966 | 580 |
| | Gender (M / F) | 82/65 | 3/1 | 94/57 | 167/95 | 67/47 |
| | Age (Mean $\pm$ Std) | $72.2 \pm 3.5$ | $73.9 \pm 4.0$ | $75.4 \pm 5.8$ | $76.6 \pm 6.9$ | $77.2 \pm 6.5$ |
| Tau | # of subjects | 30 | 37 | 48 | 31 | 20 |
| | # of records | 65 | 85 | 118 | 76 | 55 |
| | Gender (M / F) | 28/37 | 25/60 | 66/52 | 44/32 | 26/29 |
| | Age (Mean $\pm$ Std) | $68.4 \pm 3.7$ | $68.9 \pm 3.8$ | $69.8 \pm 5.9$ | $71.0 \pm 7.4$ | $78.8 \pm 8.4$ |

Table 5: Demographics of the OASIS dataset. The numbers of subjects for each category were calculated based on the baseline time point samples. The number of records denotes the total number of cross-sectional samples across the population.

| Biomarker | Category | CN | AD |
|---|---|---|---|
| Tau | # of subjects | 31 | 1 |
| | # of records | 60 | 4 |
| | Gender (M / F) | 13/18 | 1/0 |
| | Age (Mean $\pm$ Std) | $63.2 \pm 7.5$ | $71.9 \pm 1.6$ |

Table 6: Hyperparameters of ConDOR for all experiments.

| | Hyperparameter | CT | Amyloid | FDG | Tau |
|---|---|---|---|---|---|
| Train | Optimizer | Adam | Adam | Adam | Adam |
| | Learning rate | 0.001 | 0.001 | 0.0008 | 0.0008 |
| | Learning rate scheduler | MultiplicativeLR | MultiplicativeLR | MultiplicativeLR | MultiplicativeLR |
| | Learning rate decay | 0.999 | 0.999 | 0.999 | 0.999 |
| | Batch size | 16 | 16 | 16 | 16 |
| | Number of epochs | 10000 | 10000 | 10000 | 10000 |
| KDE | bandwidth $h_b$ | 0.001 | 0.001 | 0.001 | 0.001 |
| $\epsilon_\theta$ | Hidden dimension | 64 | 64 | 64 | 64 |
| | Number of layers | 4 | 4 | 4 | 4 |
| | Number of initial channel | 1 | 1 | 1 | 1 |
| | Number of hidden channels | [64, 128, 256, 512] | [64, 128, 256, 512] | [64, 128, 256, 512] | [64, 128, 256, 512] |
| | Number of final channels | 1 | 1 | 1 | 1 |
| | Number of sampling steps | 1000 | 1000 | 1000 | 1000 |
| $\epsilon_\phi$ | Hidden dimension | 64 | 64 | 64 | 64 |
| | Number of layers | 4 | 4 | 4 | 4 |
| | Number of initial channel | 1 | 1 | 1 | 1 |
| | Number of hidden channels | [64, 128, 256, 512] | [64, 128, 256, 512] | [64, 128, 256, 512] | [64, 128, 256, 512] |
| | Number of final channels | 1 | 1 | 1 | 1 |
| | Number of sampling steps | 100 | 100 | 100 | 100 |
| | cohort-level weight $\lambda$ | 0.1 | 0.1 | 0.1 | 0.7 |

## C  LIMITATION AND FUTURE WORK

**Limitations. (1)** Compared to one-shot generative methods such as TVAE and GANs, our autoregressive approach has a longer generation time since samples are generated sequentially. The gap in the sampling time becomes significantly larger as the sequence length (i.e., the number of time points) becomes extended. **(2)** Also, our method may face challenges in scenarios where labels are abruptly reversed over time. In such cases, the interpolated label $y_t^d$ may not represent a feasible disease severity and could deviate from the range of given observed labels, which may highly likely aggravate the training stability and model performance. In our experiments, we confirmed that all sequence labels are monotonically deteriorating, so the $y_t^d$ in equation 10 is defined under the strict assumption of ordinal transitions. **(3)** The four AD biomarkers used in the experiments (i.e., cortical thickness, SUVR of Amyloid, FDG, and Tau) are widely recognized and clinically validated biomarkers for identifying AD (Querbes et al., 2009; Jack & Holtzman, 2013). However, these biomarkers may not fully represent the broader spectrum of neurodegenerative diseases, such as Parkinson's or Huntington's disease, which involve different biological mechanisms. **(4)** Additionally, the AD datasets utilized in our experiments primarily consist of participants from the US, potentially limiting the model's applicability to populations with different genetic, environmental, or lifestyle factors. This lack of population diversity could affect the generalizability of the findings across broader demographics.

**Future Works. (1)** While the dual-sampling approach in our method effectively balances both cohort and subject-level information, it may introduce biased results if the trade-off hyperparameter $\lambda$ is not appropriately tuned. The ablation studies on the $\lambda$ in Table 3 indicate that the bias is not critical in general. However, considering the inefficiency of fine-tuning for every dataset, we plan to develop adaptive mechanisms that dynamically adjust the $\lambda$ based on dataset characteristics, ensuring optimal balance between cohort and subject-level contributions. **(2)** Moreover, we plan to extend our model by incorporating additional AD-relevant conditions, such as blood plasma (Schneider et al., 2009; Hansson et al., 2023). Along with the age and diagnostic labels, this biomarker could be utilized as time-dependent conditions and could provide complementary information to understand AD progression, ultimately enhancing the biological plausibility and robustness of the generated data.

# D QUALITATIVE COMPARISON WITH BASELINE MODELS

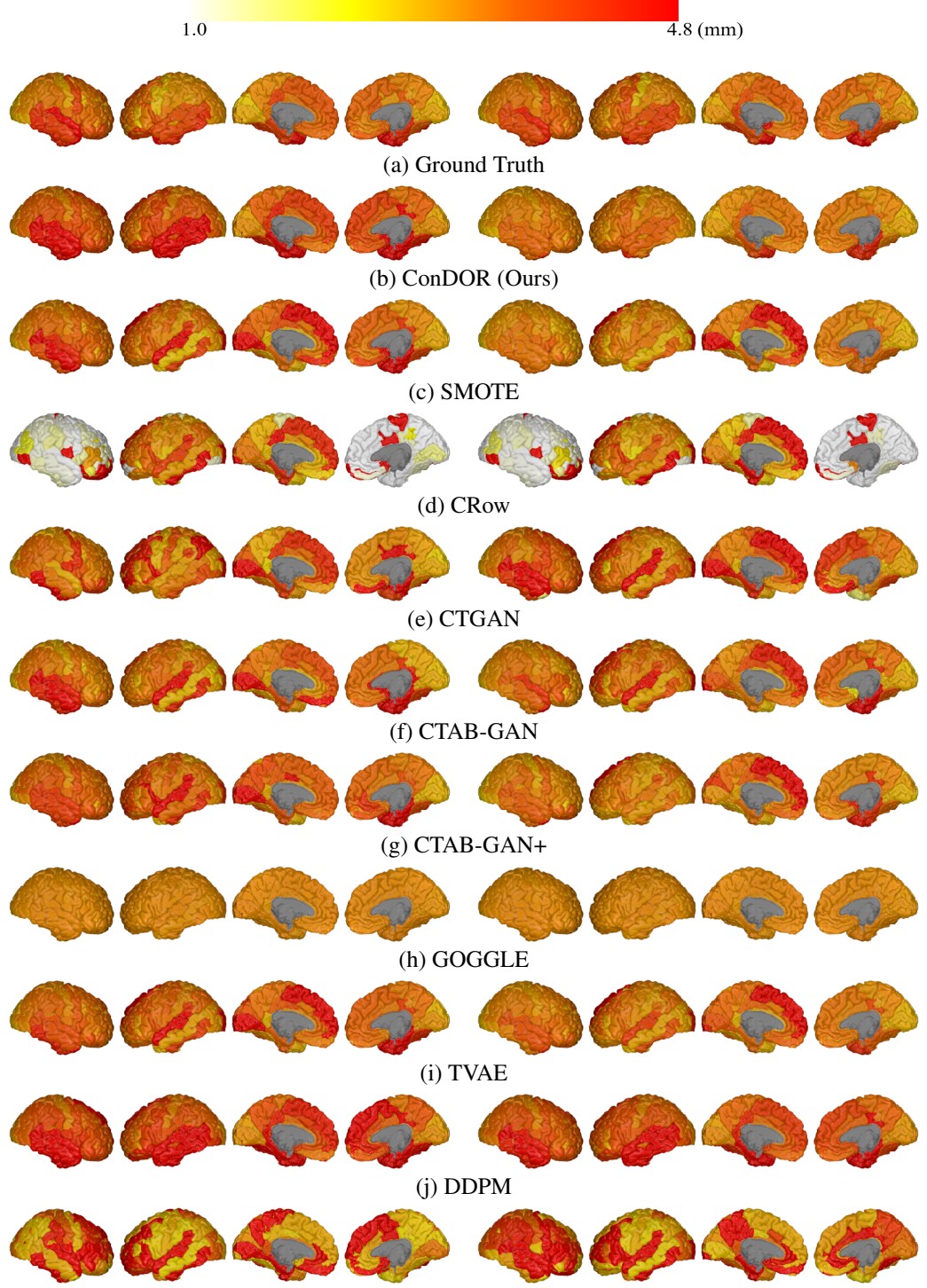

Figure 4: Example of ground truth and generated results from the CT experiment. This subject (ID: 029_S_4385) has two time points with age/label 68.2/CN at the first time point and 71.8/AD at the second time point. Each set, consisting of the outer right, outer left, inner left, and inner right hemispheres, shows the results from one time point, with time points increasing from left to right.

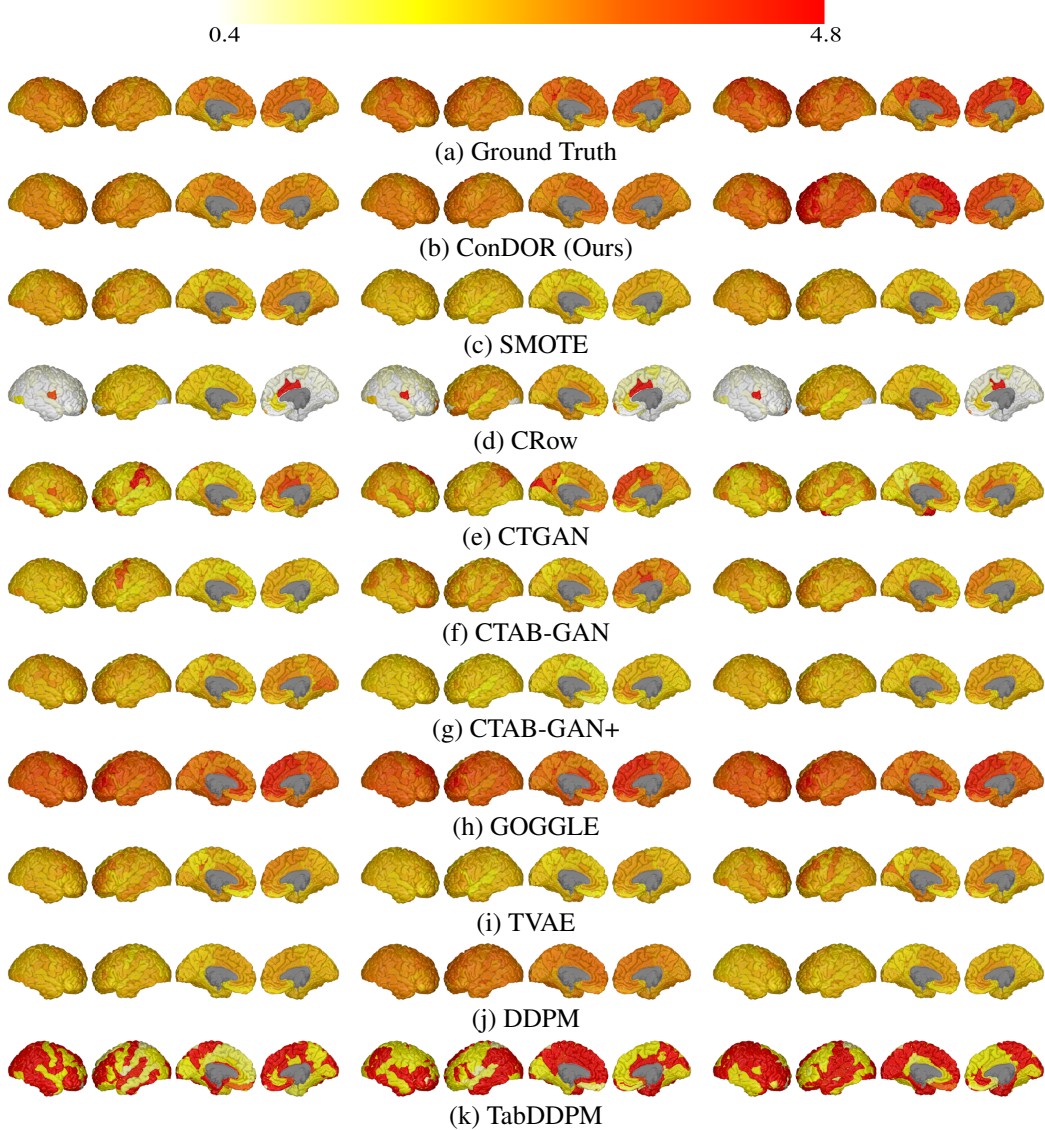

(a) Ground Truth

(b) ConDOR (Ours)

(c) SMOTE

(d) CRow

(e) CTGAN

(f) CTAB-GAN

(g) CTAB-GAN+

(h) GOGGLE

(i) TVAE

(j) DDPM

(k) TabDDPM

Figure 5: Example of ground truth and generated results from the Amyloid experiment. This subject (ID: 127_S_2213) has three time points with age/label 82.3/EMCI at the first time point, 84.3/EMCI at the second time point, and 86.3/EMCI at the third time point. Each set, consisting of the outer right, outer left, inner left, and inner right hemispheres, shows the results from one time point, with time points increasing from left to right.

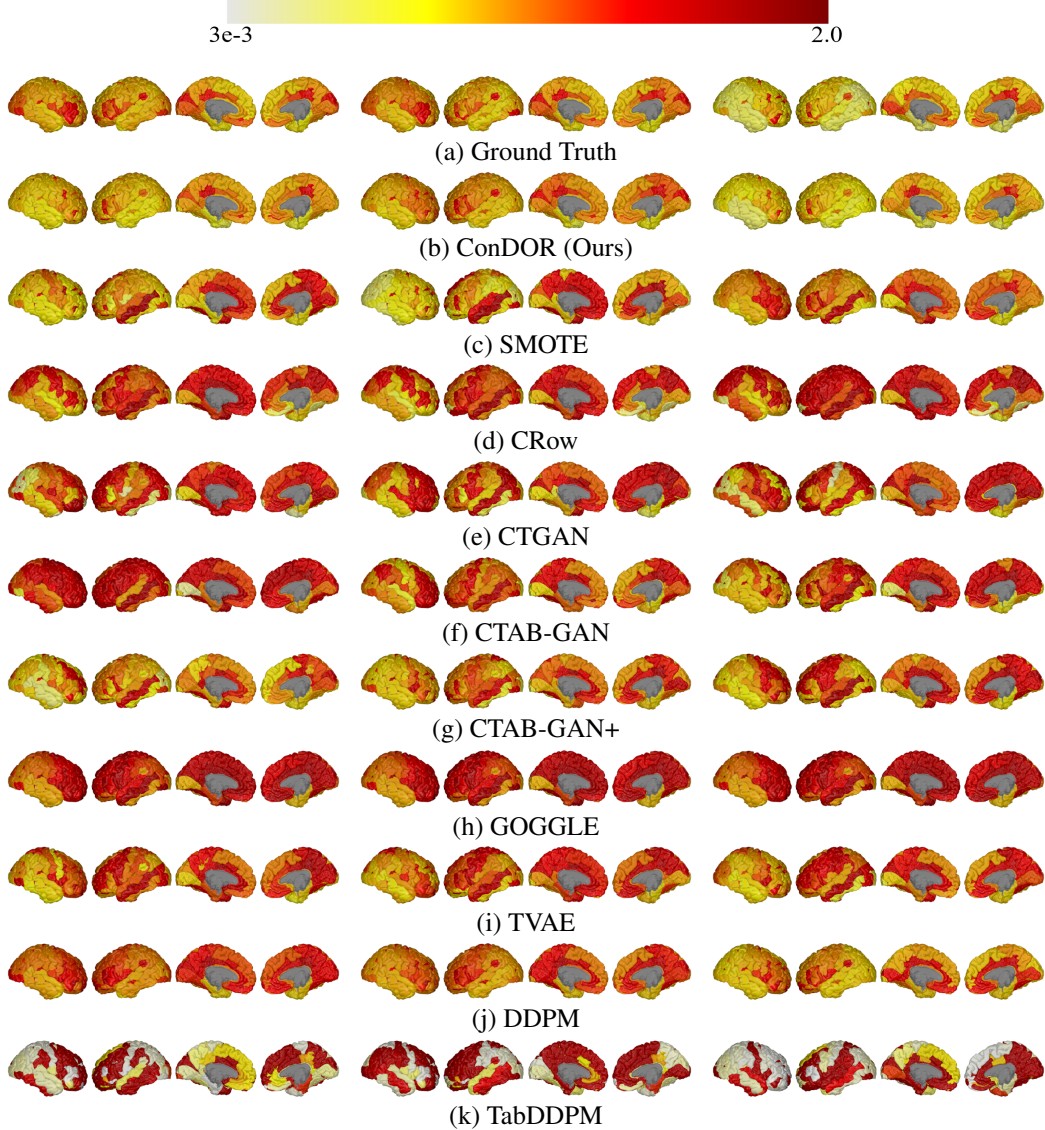

Figure 6: Example of ground truth and generated results from the FDG experiment. This subject (ID: 037_S_1078) has three time points with age/label 70.6/LMCI at the first time point, 72.3/LMCI at the second time point, and 75.6/AD at the third time point. Each set, consisting of the outer right, outer left, inner left, and inner right hemispheres, shows the results from one time point, with time points increasing from left to right.

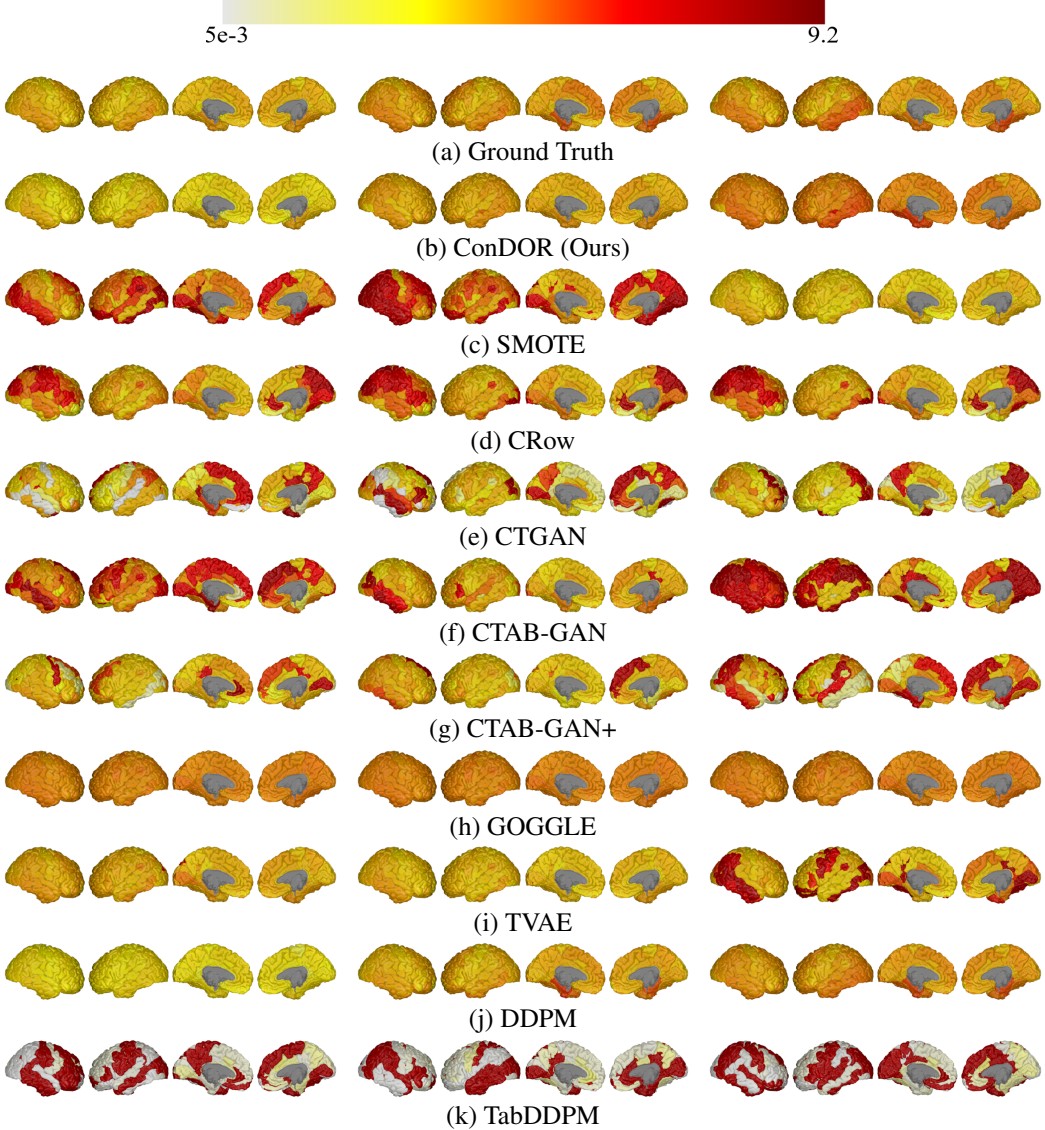

5e-3                                                                                        9.2

(a) Ground Truth

(b) ConDOR (Ours)

(c) SMOTE

(d) CRow

(e) CTGAN

(f) CTAB-GAN

(g) CTAB-GAN+

(h) GOGGLE

(i) TVAE

(j) DDPM

(k) TabDDPM

Figure 7: Example of ground truth and generated results from the ADNI dataset on Tau experiment. This subject (ID: 023_S_1190) has three time points with age/label 76.5/EMCI at the first time point, 78.0/AD at the second time point, and 79.3/AD at the third time point. Each set, consisting of the outer right, outer left, inner left, and inner right hemispheres, shows the results from one time point, with time points increasing from left to right.

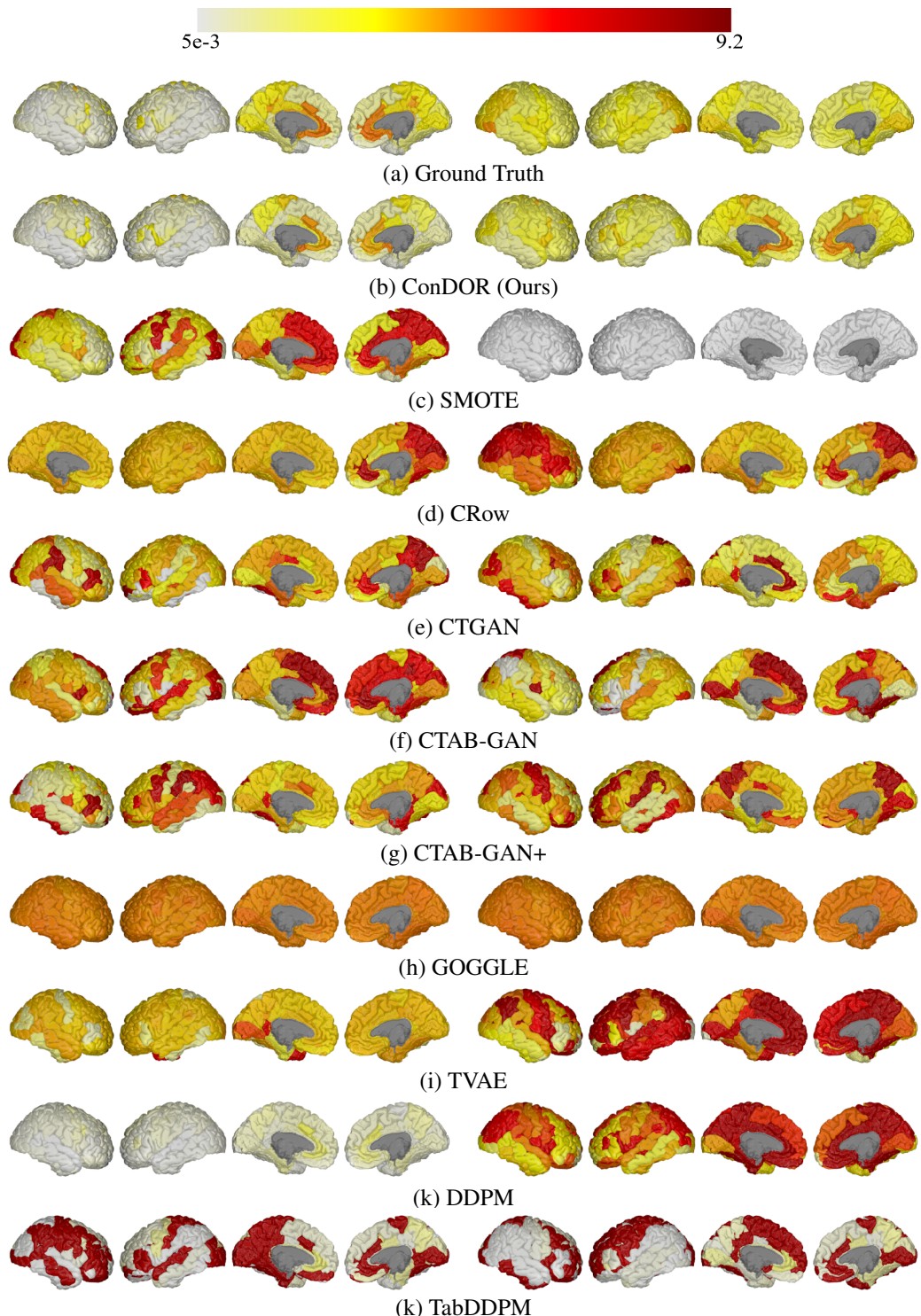

Figure 8: Example of ground truth and generated results from the OASIS dataset on the Tau experiment. This subject (ID: OAS30818) has two time points with age/label 70.0/CN at the first time point and 70.5/AD at the second time point. Each set, consisting of the outer right, outer left, inner left, and inner right hemispheres, shows the results from one time point, with time points increasing from left to right.

# E    ADDITIONAL QUALITATIVE RESULTS: A VIDEO SHOWING 20-YEAR CHANGES IN AMYLOID SUVR

In the supplementary material, we provide a video showcasing qualitative results from ConDOR on Amyloid SUVR, illustrating the sequential changes of a subject aged 65 to 85, with five diagnostic labels transitioning from CN to AD. Considering the baseline age distributions of the five labels used in the Amyloid experiment (Table 4), the age range from 65 to 85 appropriately aligns with the real-world longitudinal samples labeled from CN to AD. The video illustrates that the Amyloid SUVR of the generated sample generally increases as the disease worsens. These results are consistent with the established patterns of Amyloid accumulation observed in Alzheimer's disease, demonstrating the effectiveness of ConDOR in capturing the dynamics of the biomarker along the disease progression.

