# OpenReview forum: "Conditional Diffusion with Ordinal Regression: Longitudinal Data Generation for Neurodegenerative Disease Studies"
_ICLR.cc/2025/Conference — ICLR 2025 Spotlight_

### Official Review · Reviewer_jkdK · 2024-11-02

**Soundness:** 3
**Presentation:** 2
**Contribution:** 3
**Rating:** 8
**Confidence:** 3

**Summary:**

The paper presents ConDOR, a novel conditional diffusion model for generating longitudinal neurodegeneration data with ordinal disease progression factors. The model's architecture integrates both cohort-level and subject-level characteristics through a dual-component approach. At the cohort level, it employs Bayes' Theorem combining an ordinal regression model (capturing disease stage relationships) with a kernel-based conditional distribution. For data generation, ConDOR utilizes two diffusion models: a Regional Diffusion Model (RDM) for generating baseline measurements across brain regions, and a Temporal Diffusion Model (TDM) for generating subsequent longitudinal data. The model also incorporates a domain conditioning mechanism to integrate data from multiple sources. The authors evaluate ConDOR on multiple biomarkers (Amyloid, Cortical Thickness, and Fluorodeoxyglucose) from two prominent neurodegenerative disease datasets (ADNI and OASIS), comparing against nine baseline methods including GANs, VAEs, and other diffusion-based approaches.

**Strengths:**

1. The proposed model captures both spatial and temporal features through a combination of a Regional Diffusion Model and a Temporal Diffusion Model.
2. This new generative model addresses challenges associated with sparse, irregular, and widely spaced intervals in medical data.
3. The model strikes a balance between cohort-level and individual-level fitting, capturing generalized population trends while accommodating individual variability.
4. It introduces a novel integration of ordinal regression with diffusion models.
5. The experiments are comprehensive, with comparisons to nine baseline models, including GANs, VAEs, and other diffusion-based models, evaluated across three metrics. Implementation time is also compared.
6. The model is extended to a multi-domain setting, enhancing its generalizability and applicability to different data sources.

**Weaknesses:**

1. The ordinal regression model might oversimplify the disease progression process. Additionally, the temporal diffusion relies on linear interpolation for temporal transitions, which may not accurately capture realistic disease dynamics.
2. There is a lack of comparison with traditional longitudinal baseline models commonly used in medical literature.
3. The model evaluation has not been clearly described. Did the authors split subjects into training and test sets, keeping all observations from each subject together, or did they split individual observations, potentially placing different time points from the same subject in both training and test sets?
4. The reproducibility of this work is not guaranteed, as the code has not yet been made available.

**Questions:**

1. The model evaluation lacks clarity regarding whether the authors performed a subject-level or observation-level split. Specifically, did they keep all observations from each subject together, or did they split individual observations, potentially including different time points from the same subject in both training and test sets? It would be valuable to see how well the model predicts follow-up scans based on data from earlier time points, given that the Temporal Diffusion Model is a novel component. Additionally, for baseline models like DDPM that lack a temporal component, it would be interesting to understand how the authors utilize these models to generate follow-up scans over time.
2. The Temporal Diffusion Model uses linear interpolation to model progression in age and labels, which may not be ideal, as transitions between disease states are often abrupt or follow complex patterns. Furthermore, it would be beneficial to see theoretical proof that such linear interpolation preserves the properties of diffusion models.
3. Including some directions for future work in the conclusion would be beneficial for the research community.

**Details Of Ethics Concerns:**

No concerns

---

> ### Author Response · Authors · 2024-11-21
> **Response to Reviewer jkdK (Part 1)**
>
> We thank the reviewer for the thoughtful comments. We hope our rebuttal below addresses all your concerns and questions.
>
> W1 & Q2) The ordinal regression model and the linear interpolation may not accurately capture realistic disease dynamics.
>
> A) We acknowledge that ordinal regression and linear interpolation may not be the best methods for fully capturing real-world disease characteristics. However, given the irreversible characteristics of degenerative diseases, we believe the ordinal regression model is one of the most reasonable approaches for handling the phased and categorized nature of disease severity, which is a shared feature across the entire population.
>
> Also, although abrupt changes may occur for some individuals, such changes can be captured by linear interpolation if (1) they are observed and (2) follow the order of disease severity, as the linear interpolation is performed between every pair of the observed adjacent time points. For example, if a subject with three time points has an abrupt change at the second time point, the linear interpolation captures monotonic differences between (1) the first and second time points samples and (2) the second and third time points samples. Therefore, the linear interpolation considers the abrupt and sequential changes within individuals, thereby allowing the model to learn such changes effectively.
>
> There are some corner cases where the linear interpolation may not be suitable. For example, if abrupt changes are unobserved, identifying such hidden outliers and pinpointing their exact occurrence are challenging. Moreover, if labels are abruptly reversed over time, the interpolated label $y^d_t$ does not represent a feasible disease severity, which may highly likely aggravate the training stability. However, given that we fully utilized every observed sample and confirmed that all labels are monotonically deteriorating, we believe that linear interpolation is a practical approach in our experimental setting.
>
> W2) Lack of comparison with longitudinal studies used in the medical domain.
>
> A) We thank the reviewer for pointing out this limitation. For baseline methods, we made an effort to include as many conditional generative models as possible whose official codes are publicly available, ensuring diversity across different generative approaches (e.g., GANs, VAEs, diffusion models, normalizing flow, etc). If the reviewer has recommendations for additional baseline studies for conditional tabular data generation in the medical domain, we would be happy to review their code and add their results to our comparisons.
>
> W3 & Q1) How were subjects split?
>
> A) We performed a subject-level split, keeping all observations from each subject together. Each subject has a sequence of samples {$x_t$}$^T_{t=1}$. After preprocessing the ADNI dataset, data (i.e., sequences) from 178, 687, 678, and 166 subjects were obtained for CT, Amyloid, FDG, and Tau, respectively. For the OASIS dataset, 32 subject data were obtained after preprocessing. The stratified training/test data splits were performed with an 8:2 ratio on these preprocessed data based on the baseline time point labels.
>
> W4) Concern about reproducibility.
>
> A) We understand that implementing ConDOR from scratch may not be easy. Therefore, all codes of ConDOR and baseline methods along with their pre-trained models will be released online once the paper is accepted.
>
> Q1) How well does TDM generate follow-up scans? How DDPM was used for follow-up data generation?
>
> A) In Figures 2 and 3, we presented qualitative results demonstrating the effectiveness of TDM in generating follow-up samples based on the baseline scans. Specifically, the results at $t=2$ and $t=3$ of the second rows in Figures 2 and 3 visualize the generated follow-up samples by TDM, showing general consistency with the ground truth sequences in the first rows. These visualizations of realistic follow-up samples highlight the effectiveness of TDM in capturing temporal dynamics while accounting for the initial characteristics of the baseline sample.
>
> Since some baseline generative methods, such as DDPM, are not inherently designed for longitudinal data generation, they handle temporal dynamics differently during the sampling process. Specifically, while our model takes a sequence of conditions (i.e., ages  {$a_t$}$^T_{t=1}$ and labels {$y_t$}$^T_{t=1}$) to generate a sequence of samples, baseline methods like DDPM process each time point independently. For example, DDPM takes the same cross-sectional conditions (i.e., age $a_t$ and label $y_t$) repeatedly for $T$ iterations. These $T$ independently generated data points are then aggregated to form a sequence, and the evaluation is performed based on these resulting sequences.

---

> ### Author Response · Authors · 2024-11-21
> **Response to Reviewer jkdK (Part 2)**
>
> Q2) Does linear interpolation preserve the properties of diffusion models?
>
> A) Given that the linear interpolation between two adjacent time points {$x_{t-1}, x_t$} yields an interim sequence of pseudo-samples {$x_t^1, x_t^2, …, x_t^D$} with small incremental changes $\Delta x^d_t$, estimating these tiny differences aligns with the principles of a diffusion model, which inherently estimates small noises at each step of the diffusion process. These interpolated pseudo-samples serve as intermediate noisy samples for stepwise diffusion without altering the fundamental structure of the diffusion process itself.
> Notably, in conventional conditional diffusion models, conditioning variables are typically static and not directly paired with the noisy samples generated during the forward diffusion process. In contrast, our method introduces a novel conditioning mechanism where the interim pseudo-samples are explicitly paired with linearly interpolated time-dependent conditions (e.g., age $a^d_t$ and diagnostic label $y^d_t$). This pairing allows the model to explicitly learn temporal dynamics while maintaining the progressive property of conditions, ensuring consistency with the overall design of the diffusion framework. Overall, the linear interpolations on the samples and conditions preserve the key properties of diffusion models while introducing an innovative mechanism to handle longitudinal neurodegenerative data generation.
>
> Q3) Adding future works would be beneficial.
>
> A) Thank you for suggesting this valuable addition. We added the ‘Limitation and Future Work’ section in Appendix C, so please refer to it for a detailed discussion of potential improvements and future directions. We hope this addition addresses the reviewer’s concerns and contributes to future advancements in medical data analysis.

---

> > ### Comment · Reviewer_jkdK · 2024-12-02
> > **Further response**
> >
> > I appreciate the responses from the authors and would like to increase the score.

---

### Official Review · Reviewer_xa3Q · 2024-11-03

**Soundness:** 2
**Presentation:** 2
**Contribution:** 3
**Rating:** 6
**Confidence:** 4

**Summary:**

This paper proposes a conditional generative model for synthesizing longitudinal sequences.
It first uses ordinal regression and kernel density estimation to model the conditional PDF and then interpolate gaps between consecutive observations.
Two diffusion models are then trained to model baseline data and changes in follow-up samples.
These diffusion models generate disease progression data by sequentially sampling.

**Strengths:**

The design of the proposed model's structure design is innovative, partsuch as the combination of two diffusion models and the use of both cohort and subject-level interpolation for training.

This paper provides clear comparisons to baseline models across multiple metrics and good visualizations.

**Weaknesses:**

Some abnormal results are not well discussed or explained.
For example, in Table 1, the variance in DDPM's performance is very high compared to other methods.
Since DDPM is one of only two diffusion-based comparison methods, it would be helpful to provide an explanation for this abnormal performance.


The authors do not mention how to deploy the model. For example, how to use the trained model to generate new data.
If my understanding of the generative process is correct, we only need to use these two diffusion models with random noise as the input when generating new data.
There is no option to allow the two diffusion models to generate samples conditioned on specific ages and disease severity,
e.g., we can't use age as an input for these diffusion models when generating,
although the authors claim that this model can generate data conditioned on these factors.

Some important technical details are also missing. Please refer to the questions below

**Questions:**

RDM is designed to geneate the baseline sample $x_1$. However, during training, the authors use all the samples $x_t$ (t = 1, .., t) as independent cross-sectional data.
The justification for using such setting is not presented in the paper.
For example, why not use only $x_1$ to train the RDM?
Will this setting cause the RDM to be biased as some longitudinal samples have longer records or are recorded more frequently

Acoording to Table 3, the choice of hyperparameter $\lambda$ significantly impacts the performance of the proposed method.
Howerver, how to select $\lambda$ is unclear, for example, which dataset and what metric do the author use to choose $\lambda$.
The author used 80% of the whole data for training and the rest 20% for testing for all experiments,
and it seems there is no validation dataset to optimize $\lambda$.
Therefore, the results shown in Table 3 is less convincing to me since they are propbably derived from either training or test set,
and the results in Table 1 and 2 are also less convincing since we may not able to get the best $\lambda$ in practice.


The training strategies for the generative model are somewhat unclear to me. For example do the authors train RDM and TDM seperatedly or jointly?
$D$ seems to be another important hyperparameter, but how the authors chose $D$ is also unclear.

---

> ### Author Response · Authors · 2024-11-21
> **Response to Reviewer xa3Q (Part 1)**
>
> We thank the reviewer for your time and thoughtful comments. We address all concerns raised by the reviewer, and we hope that the reviewer reconsiders the score favorably towards acceptance.
>
> W1) Why are the variances of DDPM results higher than those of other baselines?
>
> A) We acknowledge that the reported standard deviations of DDPM are generally higher compared to other methods, as shown in Table 1. This higher standard deviation likely arises from the sensitivity of DDPM to hyperparameter tuning, particularly the learning rate and parameter initialization, which can impact its convergence behavior. To find the optimal hyperparameters for DDPM, we performed a grid search to select the optimal learning rate of DDPM from {0.002, 0.015, 0.001, 0.0008, 0.0005} as we did for ConDOR. For each learning rate, we ran three independent trainings from scratch with different parameter initializations and reported the best average results.
>
> While the reported mean represents the best overall performance, the accompanying standard deviation reflects the variability across these multiple training runs at the optimal learning rate (0.0005). In contrast, for other learning rates, the standard deviation was considerably smaller, but the mean performance was not as strong. For example, in the Amyloid experiment with learning rate=0.001, the DDPM results were 17.69 ($\pm$ 1.199), 1.02 ($\pm$ 0.050), and 0.07 ($\pm$ 0.009) for WD, RMSE, and JSD, respectively, showing significantly lower standard deviations but poorer mean performance compared to the optimal setup.
>
> Moreover, the standard deviation can be affected by metrics and datasets. For example, in the FDG experiment, the standard deviation of TVAE on RMSE (0.019) and GOGGLE on JSD (0.002) were higher than those of DDPM. Also, in the Tau experiment, CRow and CTGAN showed 3.2 and 1.8 times larger standard deviation on JSD than that of DDPM. These results suggest that other baseline methods can also have high standard deviations depending on the selected metrics and data characteristics.
>
> W2) The authors did not mention how to use the model for sampling new data. The proposed model cannot use conditions during sampling.
>
> A) We are sorry but we think there is a misunderstanding. In the sampling process, we used a trained $\textbf{conditional}$ U-Net ($\mu_{\theta}$) to generate unseen samples. In Eq. 8 of the original manuscript (in line 229), we mentioned that the $\mu_{\theta}(x^d_t, a_t, y_t, d)$ takes conditions (i.e., age $a_t$ and disease severity $y_t$). Using a conditional U-Net to implement a conditional diffusion model is the convention [1], and we followed this existing method for conditional sample generation. All quantitative and qualitative results in the paper were derived from the trained $\mu_{\theta}(x^d_t, a_t, y_t, d)$. To enhance clarity, we revised line 233 by explicitly mentioning ‘conditional U-Net’ instead of ‘U-Net’.
>
> [1] Rombach et al, “High-Resolution Image Synthesis With Latent Diffusion Models”, CVPR 2022
>
> Q1) Why $x_t (t=1, …, T)$ were used for training RDM? Will this setting cause the RDM to be biased?
>
> A) To secure sufficient training data, we used all $t=1,...,T$ to train the RDM although the RDM aims to generate baseline time point samples. If only $x_1$ samples were used for training, only 142, 549, 542, and 132 training samples could be used from the ADNI cortical thickness, Amyloid, FDG, and Tau datasets, respectively, and 25 samples could be used for the OASIS dataset. However, if all time point samples are used, at least double samples are secured for training the RDM.
> Regarding the concern about biased training, we had the same question at the initial data preprocessing stage. Therefore, we investigated the number of time points of all subjects and assumed that the effect of bias would be marginal as most subjects have two time points (70%) or three time points (20%) samples.

---

> ### Author Response · Authors · 2024-11-21
> **Response to Reviewer xa3Q (Part 2)**
>
> Q2) How to choose $\lambda$? No validation set to optimize $\lambda$.
>
> A)  As shown in Table 3, we performed a grid search to find optimal $\lambda$ in {0.1, 0.3, 0.5, 0.7, 0.9} for all datasets. The best $\lambda$ for each dataset was chosen based on the smallest Wasserstein Distance (WD). Upon closer inspection of the results in Table 3, we believe that the reviewer can easily see that the overall performance differences are marginal depending on the choice of $\lambda$. While slight variations exist, the results of the smallest (0.1) and largest (0.9) $\lambda$ still consistently outperform all baseline methods in the single-domain learning experiments.
>
> We acknowledge the reviewer’s concern about not using the validation set, as we had the same consideration before conducting experiments. However, due to the limited size of data (e.g., around 150 ~ 700 data are available), we prioritized maximizing the training data by splitting the whole data into an 8:2 ratio for training and testing. To mitigate the potential for biased results, we trained our model and all baseline models 3 times from scratch on the training set and evaluated each on the test set, and the final results of each method are averaged across the 3 replicates. This experimental setting with 8:2 data split and reporting the average performance of three runs is well-established in the literature on generative methods with small samples (100 ~ 600 samples). For example, recent studies on generative models such as [2] adopt the same experimental setups to ensure model stability and reliability in small-sample scenarios.
>
> [2] Jo et al, “Score-based Generative Modeling of Graphs via the System of Stochastic Differential Equations”, ICML 2022
>
> Q3) Are RDM and TDM trained separately or jointly? How was $D$ chosen?
>
> A) The RDM and TDM are trained separately by using separate loss functions $L_\text{RDM}$ and $L_\text{TDM}$, respectively. After training both models, RDM first generates baseline samples in the sampling process. These generated baseline samples are inputted to TDM and TDM yields follow-up samples by estimating the difference between the baseline and follow-up samples.
>
> Regarding the selection of $D$, we clarify the rationale for choosing $D$ through this rebuttal, while all hyperparameter settings including $D$ are provided in Table 6. For the RDM, $D$ was set to 1000, following the number of diffusion steps used in DDPM (Ho et al., NeurIPS 2020). In contrast, $D$ was set to 100 for the TDM due to the different relationship between noise and $D$. Unlike RDM whose noise $\epsilon^d_t \sim N(0, 1)$ is independent of the diffusion step $D$, the $\epsilon_\phi \approx \Delta x_t^d$ of TDM highly depends on the $D$ as a larger $D$ results in smaller $\Delta x_t^d$. Due to this dependency, the large $D$ (i.e., small $\Delta x_t^d$) of TDM leads to a gradient explosion as the $\epsilon_\phi$ has to estimate excessively tiny values that are close to zero. To address this issue, we progressively reduced the $D$ of TDM until the training stabilized, and empirically observed that $D=100$ was sufficient to avoid the gradient explosion of TDM.

---

> > ### Comment · Reviewer_xa3Q · 2024-12-02
> >
> > Thank you for answering my questions and I have increased my score.

---

### Official Review · Reviewer_rUXZ · 2024-11-04

**Soundness:** 3
**Presentation:** 4
**Contribution:** 3
**Rating:** 8
**Confidence:** 4

**Summary:**

This paper introduces a novel conditional generative model for synthesizing longitudinal sequences to study neurodegenerative diseases such as Alzheimer’s disease. The method uses ordinal regression and a diffusion model to generate realistic disease progression imaging data.

**Strengths:**

1. Combing the cohort-level trend and subject-level trend for longitudinal data generation.
2. Extensive validation on four Alzheimer's Disease biomarkers demonstrates the model's superiority over nine baseline approaches.

**Weaknesses:**

1. The paper lacks a comprehensive theoretical justification for the proposed method. While the method is innovative, a deeper theoretical comparison with existing models could strengthen the argument for its necessity and effectiveness.
2. The description of the methodology, particularly the integration of cohort-level and subject-level samples, is somewhat convoluted. The paper could benefit from clearer explanations and more detailed algorithmic steps to enhance reproducibility.
3. The paper does not provide a thorough statistical analysis to support these claims. The lack of confidence intervals or significance testing weakens the robustness of the reported findings.
4. The discussion section is relatively weak in terms of interpreting the results and their implications. The paper does not adequately address the potential limitations of the proposed method or suggest directions for future research, which are crucial for a comprehensive understanding of the study’s impact.

**Questions:**

1. How does the proposed method theoretically ensure the accurate representation of disease progression, especially considering the complex dynamics and irregular intervals in longitudinal data?
2. How does the proposed method theoretically improve upon existing generative models for longitudinal data? Are there any theoretical limitations or assumptions that need further clarification?
3. The paper introduces a dual-sampling approach combining cohort-level and subject-level samples. How does this method compare to other state-of-the-art techniques in terms of capturing individual-specific features and general trends? Are there any potential biases introduced by this approach?
4. The experiments are conducted on four AD biomarkers from MRI and PET images. How representative are these biomarkers and datasets of the broader neurodegenerative disease population? Are there any limitations in the experimental design that could affect the generalizability of the results?
5. The paper claims superiority over nine baseline approaches. How robust are these results across different metrics and datasets? Are there any specific scenarios or conditions under which the proposed method might underperform or fail?

---

> ### Author Response · Authors · 2024-11-21
> **Response to Reviewer rUXZ (Part 1)**
>
> We thank the reviewer for your effort in reviewing the manuscript and highly valuing our research.
>
> W1 & Q2 & Q3) Theoretical comparison of the proposed method with existing generative models. Comparison of the dual-sampling method with baseline models.
>
> A) (1) To the best of our knowledge, existing studies for longitudinal degenerative disease analyses consider only observed conditions at static time points, which may fail to represent the irregular intervals and irreversible characteristics of neurodegenerative data. However, our method introduces fine-grained, interpolated time-dependent conditions (e.g., age $a^d_t$ and diagnostic labels $y^d_t$) that dynamically adapt to the irregular sampling of time points.
>
> (2) Also, we introduced a novel method of incorporating the ordinal regression model into the diffusion model, which allows the model to effectively handle the ordinality of conditions inherent in the data. These mechanisms make the model robust in dealing with temporal complexity and generating realistic and biologically plausible disease trajectories.
>
> (3) Moreover, unlike existing methods that primarily model either global trends (cohort-level information) or individual-specific features (subject-level information), our dual-sampling approach explicitly integrates both. This design ensures that the model captures the nuanced interplay between shared population-level dynamics and unique individual trajectories, which is critical for representing longitudinal data.
>
> Specifically, the dual-sampling approach in our method is designed to $\textit{balance}$ individual-specific features and general trends within longitudinal neurodegenerative data. Unlike our method, the nine baseline approaches (e.g., CTGAN, TabDDPM, SMOTE, etc) we used in our comparisons do not explicitly separate individual features from the global-level features during their feature extraction or generation processes. Instead, these methods primarily focus on either global distribution modeling or sample-level characteristics, which may limit their ability to represent the nuanced interplay between individual variability and population-level patterns. For example, CTGAN and TabDDPM focus on balancing multimodal distributions within tabular data (i.e., brain regional features in our experiments) across the whole dataset, without considering personalized variability. On the other hand, SMOTE considers only sample-wise features during generation as it synthesizes new data by combining a real data point and its $k$-nearest neighbors.
>
> W2) More detailed descriptions of the dual-sampling method are needed.
>
> A) We thank the reviewer for pointing out this aspect. We acknowledge that the description of the integration of the cohort-level and subject-level samples in Section 2.3.2 could benefit from further details. Due to the page limit, it was challenging to provide a more extensive explanation in the main text without compromising other critical aspects of the paper. However, to address this, we plan to make the official code publicly available, including detailed documentation and step-by-step instructions to ensure reproducibility.
>
> W3) Lack of statistical analysis (e.g., confidence intervals or significance testing).
>
> A) Thank you for the suggestion. While we have reported the means and standard deviations of all results from multiple runs to demonstrate model generalizability, we understand that adding additional statistical analyses could further enhance the robustness of the results. However, we had difficulty in determining which specific statistical tests would be most appropriate to further analyze the results, so we would appreciate any specific recommendations from the reviewer regarding statistical techniques that could best address this concern.

---

> ### Author Response · Authors · 2024-11-21
> **Response to Reviewer rUXZ (Part 2)**
>
> W4 & Q5) Lack of discussion of limitations and future works. Are there any specific scenarios under which the proposed method might underperform?
>
> A) Thank you for your valuable feedback. We added the ‘Limitation and Future Work’ section in Appendix C, so please refer to it for a detailed discussion of potential improvements and future directions. We hope this addition addresses the reviewer’s concerns and contributes to advancing medical data analysis.
>
> As outlined in the appendix, we acknowledge several weaknesses and scenarios where our method might underperform. For example, compared to one-shot generative methods such as TVAE and GANs, our autoregressive approach has a longer generation time since samples are generated sequentially. The gap in the sampling time becomes significantly larger as the sequence length (i.e., the number of time points) becomes extended.
>
> Additionally, our method may face challenges in scenarios where labels are abruptly reversed over time. In such cases, the interpolated label $y^d_t$ may not represent a feasible disease severity and could deviate from the range of given observed labels, which may highly likely aggravate the training stability and model performance. In our experiments, we confirmed that all sequence labels are monotonically deteriorating, so that the $y^d_t$ in Eq. 10 is defined under the strict assumption of ordinal transitions.
>
> Q1) How does ConDOR theoretically ensure the accurate representation of disease progression, considering complex dynamics and irregular intervals?
>
> A) By using fine-grained and interpolated time-dependent conditions (i.e., age $a^d_t$ and diagnostic label $y^d_t$) for each diffusion step $d=1, ..., D$ and time point $t=1, ..., T$, our model effectively captures realistic disease progression with complex dynamics and irregular intervals. These conditions, defined at unobserved time points (i.e., at diffusion step $d$ between observed $t-1$ and $t$), enable the method to estimate unobserved sample points from the conditional PDF $f_{X|A, Y}(x_t|a_t, y_t)$. For example, consider a subject with two time points where labels change from Cognitively Normal (CN) to Alzheimer’s Disease (AD). Existing baseline methods only utilize these two observed labels, overlooking the gradual transition of diagnostic labels between them. In contrast, our method accounts for all intermediate labels, reflecting the irreversible and progressive deterioration from CN to AD over time. Additionally, our method incorporates intermediate ages spanning the interval from $t-1$ to $t$. This mechanism allows our model to capture biologically plausible disease trajectories across arbitrary intervals, addressing the limitations of existing methods in handling complex and irregular temporal dynamics of neurodegenerative diseases.
>
> Q3) Any potential biases introduced by the dual-sampling method?
>
> A) While the dual-sampling approach in our method effectively balances both cohort and subject-level information, it may introduce biased results if the trade-off hyperparameter $\lambda$ is not appropriately tuned. The ablation studies on the $\lambda$ in Table 3 indicate that the bias is not critical in general; however, considering the inefficiency of fine-tuning for every dataset, we plan to develop adaptive mechanisms that dynamically adjust the $\lambda$ based on dataset characteristics, ensuring optimal balance between cohort and subject-level contributions.
>
> Q4) How representative are the four biomarkers for a broader disease population? Any limitations in the experimental design?
>
> A) The four AD biomarkers used in the experiments (i.e., cortical thickness, SUVR of Amyloid, FDG, and Tau) are widely recognized and clinically validated biomarkers for identifying AD [1, 2]. However, these biomarkers may not fully represent the broader spectrum of neurodegenerative diseases, such as Parkinson’s or Huntington’s disease, which involve different biological mechanisms. Additionally, the AD datasets utilized in our experiments primarily consist of participants from the US, potentially limiting the model’s applicability to populations with different genetic, environmental, or lifestyle factors. This lack of population diversity could affect the generalizability of the findings across broader demographics.
>
> [1] Querbes et al. "Early diagnosis of Alzheimer's disease using cortical thickness: impact of cognitive reserve." Brain, 2009
>
> [2] Jack et al, "Biomarker modeling of Alzheimer’s disease." Neuron, 2013

---

### Author Response · Authors · 2024-11-21
**General response to all reviewers**

We sincerely thank all reviewers for their thoughtful and constructive feedback. We have addressed all concerns raised by the reviewers and kindly request that the reviewers consider our rebuttals.

Also, we appreciate the recognition of our contributions, practical value, and experimental rigor. Notably, the reviewers highlighted:

$\textbf{(1) Technical Novelty}$:

$\bullet$ The design of the proposed model's structure design is innovative, e.g., the combination of two diffusion models and the use of both cohort and subject-level interpolation for training. ($\textbf{reviewer xa3Q}$)

$\bullet$ This new generative model addresses challenges associated with sparse, irregular, and widely spaced intervals in medical data. ($\textbf{reviewer jkdK}$)

$\bullet$ It introduces a novel integration of ordinal regression with diffusion models. ($\textbf{reviewer jkdK}$)

$\bullet$ This paper introduces a novel conditional generative model for synthesizing longitudinal sequences to study neurodegenerative diseases such as Alzheimer’s disease. ($\textbf{reviewer rUXZ}$)

$\textbf{(2)  Model Strengths and Capabilities}$:

$\bullet$ The proposed model captures both spatial and temporal features through a combination of a Regional Diffusion Model and a Temporal Diffusion Model. ($\textbf{reviewer jkdK}$)

$\bullet$ The proposed model combines the cohort-level trend and subject-level trend for longitudinal data generation. ($\textbf{reviewer rUXZ}$)

$\bullet$ The model strikes a balance between cohort-level and individual-level fitting, capturing generalized population trends while accommodating individual variability. ($\textbf{reviewer jkdK}$)

$\bullet$ The model is extended to a multi-domain setting, enhancing its generalizability and applicability to different data sources. ($\textbf{reviewer jkdK}$)

$\textbf{(3) Extensive Experiments}$:

$\bullet$ Extensive validation on four Alzheimer's Disease biomarkers demonstrates the model's superiority over nine baseline approaches. ($\textbf{reviewer rUXZ}$)

$\bullet$ This paper provides clear comparisons to baseline models across multiple metrics and good visualizations. ($\textbf{reviewer xa3Q}$)

$\bullet$ The experiments are comprehensive, with comparisons to nine baseline models, including GANs, VAEs, and other diffusion-based models, evaluated across three metrics. Implementation time is also compared. ($\textbf{reviewer jkdK}$)

---

> ### Author Response · Authors · 2024-11-27
>
> Dear Reviewers,
>
> We hope that our rebuttal has addressed your concerns and now provides a clearer understanding of our work based on your reviews.
> We appreciate your comments, and should you have any additional questions or require further clarification, please do not hesitate to let us know.
> We hope to contribute to the ICLR and broader ML/Neuroscience community by having this paper published.

---

### Public Comment · ~Hyuna_Cho1 · 2025-03-18
**Code Release**

We share the code of ConDOR and all baseline models we used in our study, along with the pretrained models for all experimental settings, in this repository: https://github.com/Hannah37/ConDOR-ICLR25/tree/main

---

### Meta-Review · Area_Chair_LSRm · 2024-12-19

**Metareview:**

The paper proposes ConDOR, a novel conditional generative model for synthesizing longitudinal sequences and present its application to neurodegenerative disease data generation conditioned on multiple time-dependent ordinal factors, such as age and disease severity. The synthetic data are curated to integrate both cohort-level and individual-specific characteristics, where the cohort-level representations are modeled with an ordinal regression to capture longitudinally monotonic behavior. Extensive experiments are conducted. In sum: the algorithm is innovative, the problem comes from real-world challenges from the broader ML/Neuroscience community, and the experimental results are convincing. After the rebuttal stage, all the reviewers have unanimously supported this paper.

PC/SAC/AC and all reviewers will monitor this submission to ensure the reproductivity of this work - the authors promised to release codes of their methods and baseline methods along with their pre-trained models publicly. Raw data and/or descriptions of how to access the data in Appendix A are also required by the ICLR community.

**Additional Comments On Reviewer Discussion:**

The authors have sufficiently addressed most comments from all the reviewers. Reviewers have increased their scores accordingly during the discussions.

---

### Decision · Program_Chairs · 2025-01-22

Accept (Spotlight)